# Behaviour change interventions improve maternal and child nutrition in sub-Saharan Africa: A systematic review

**Daniella Watson**[1,2,3]*, **Patience Mushamiri**[4], **Paula Beeri**[5], **Toussaint Rouamba**[6], **Sarah Jenner**[7], **Simone Proebstl**[7,8,9], **Sarah H Kehoe**[1,7], **Kate A Ward**[1,7,10], **Mary Barker**[7,10,11,12], **Wendy Lawrence**[7,11], **the INPreP Study Group**[¶]

1 Global Health Research Institute, Human Development and Health, Faculty of Medicine, University of Southampton, Southampton, United Kingdom, 2 Department of Global Health and Social Medicine, King's College London, London, United Kingdom, 3 SAMRC Developmental Pathways for Health Research Unit, School of Clinical Medicine, University of the Witwatersrand, Johannesburg, South Africa, 4 SAMRC Centre for Health Economics and Decision Science, PRICELESS, University of the Witwatersrand, School of Public Health, Faculty of Health Sciences, Johannesburg, South Africa, 5 Navrongo Health Research Centre, Ghana Health Service, Accra, Ghana, 6 Clinical Research Unit of Nanoro, Institute for Research in Health Sciences, National Center for Scientific and Technological Research, Ouagadougou, Burkina Faso, 7 Medical Research Council Lifecourse Epidemiology Centre, University of Southampton, Southampton, United Kingdom, 8 Institute for Medical Information Processing, Biometry, and Epidemiology—IBE, LMU Munich, Munich, Germany, 9 Pettenkofer School of Public Health, Munich, Germany, 10 School of Public Health, Faculty of Health Sciences, University of the Witwatersrand, Johannesburg, South Africa, 11 NIHR Southampton Biomedical Research Centre, University Hospitals Southampton NHS Foundation Trust, Cambridge, United Kingdom, 12 School of Health Sciences, Faculty of Life and Environmental Sciences, University of Southampton, Southampton, United Kingdom

¶ Membership of INPreP Study Group is provided in the Acknowledgments.
* daniella.watson@kcl.ac.uk

**Data Availability Statement:** This is secondary data and can be accessed through the original research articles.

## Abstract

Evidence that nutrition-specific and nutrition-sensitive interventions can improve maternal and child nutrition status in sub-Saharan Africa is inconclusive. Using behaviour change theory and techniques in intervention design may increase effectiveness and make outcomes more predictable. This systematic review aimed to determine whether interventions that included behaviour change functions were effective. Six databases were searched systematically, using MeSH and free-text terms, for articles describing nutrition-specific and nutrition-sensitive behaviour change interventions published in English until January 2022. Titles, abstracts and full-text papers were double-screened. Data extraction and quality assessments followed Centre for Reviews and Dissemination guidelines. Behaviour change functions of interventions were mapped onto the COM-B model and Behaviour Change Wheel. PROSPERO registered (135054). The search yielded 1193 articles: 79 articles met inclusion criteria, ranging from low (n = 30) to high (n = 11) risk of bias. Many that applied behaviour change theory, communication or counselling resulted in significant improvements in infant stunting and wasting, household dietary intake and maternal psychosocial measures. Interventions with >2 behaviour change functions (including persuasion, incentivisation, environmental restructuring) were the most effective. We recommend incorporating behaviour change functions in nutrition interventions to improve maternal and child outcomes, specifically drawing on the Behaviour Change Wheel, COM-B model (SORT B

**Funding:** This research was funded by the National Institute for Health Research (NIHR) (17\63\154) using UK aid from the UK Government to support global health research (DW, PB, TR, SHK, KAW, MB). The views expressed in this publication are those of the authors and not necessarily those of the NIHR or the UK Department of Health and Social Care. The funders had no role in study design, data collection and analysis, decision to publish, or preparation of the manuscript.

**Competing interests:** The authors have declared that no competing interests exist.

recommendation). To enhance the designs of these interventions, and ultimately improve the nutritional and psychosocial outcomes for mothers and infants in sub-Saharan Africa, collaborations are recommended between behaviour change and nutrition experts, intervention designers, policy makers and commissioners to fund and roll-out multicomponent behaviour change interventions.

## Background

The triple burden of malnutrition [1], coexistence of under- and over-nutrition [2] and poor micronutrient status impacts maternal and child health [3–6]. In sub-Saharan Africa this occurs at the household level, where family members can be underweight (BMI$<18.5$kg/m$^2$) [7] or overweight (BMI$>25$kg/m$^2$) [7], and at the individual level, where an infant can be stunted (height for age $<$ -2 SD from the median) [7], wasted (weight for age $<$-2SD from the median) [7] or overweight [8]. One meta-analysis found that rural residency, low educational status of partners, multiple pregnancies and poor nutritional indicators were determinants of malnutrition in pregnancy [9]. A recent Lancet series [3–6, 10–17] advocated for person-centred health systems and proposed a lifecourse double-duty approach to optimise dietary quality to address both under- and over-nutrition [11–14]. They advise using both "nutrition-specific" and "nutrition-sensitive" interventions [3–6]. Large-scale nutrition-sensitive interventions (agriculture, clean water, sanitation, education and employment) may enhance nutrition-specific interventions (breastfeeding promotion, fortification of foods, and micronutrient supplementation) by addressing underlying determinants of nutrition [5, 18]. Approaches to improve infant nutrition status include 1) promotion of individual behaviour change to address diet quality, 2) food fortification, 3) nutritional supplementation [18–20].

Behavioural functions of interventions are increasingly recognised as crucial, yet expertise in global health behaviour change is lacking [21]. Shelton (2013) described behaviour change as the missing block in global health systems, pointing out that the top 20 health risks (e.g. obesity, childhood underweight, vitamin deficiencies) in sub-Saharan Africa are influenced by behaviour (e.g. care seeking, adherence, pro-social behaviours) [21]. Public health and health promotion interventions based on social and behavioural science theories are more effective than interventions without such a theoretical base [22]. Further, a lack of theory-informed evaluations limits what can be learnt about how interventions work in different contexts, how health impacts are observed, and effects on primary and secondary outcomes [23–25].

Research suggests that intervention strategies must combine behaviour change with access to nutritious food [26], acknowledging that people eat 'food' rather than 'nutrients' [27]. Behavioural scientists and health psychologists argue that those designing behaviour change interventions should engage with the target population, understand their motivation to change, and adapt interventions to the contexts that facilitate change including environment and social networks [26]. One model used by Health Psychologists is the Behaviour Change Wheel (Fig 1) which encapsulates features applicable to intervention design and implementation [28]. It outlines nine behavioural intervention functions, that aims to address deficits in one or more of the three underlying human factors–**C**apability (physical and psychological), **O**pportunity (physical and social), and **M**otivation (automatic and reflective) that influence **B**ehaviour (COM-B model) (Fig 2). Both models can be applied to inform the design of interventions to improve maternal and child nutrition in the context into which they are being implemented.

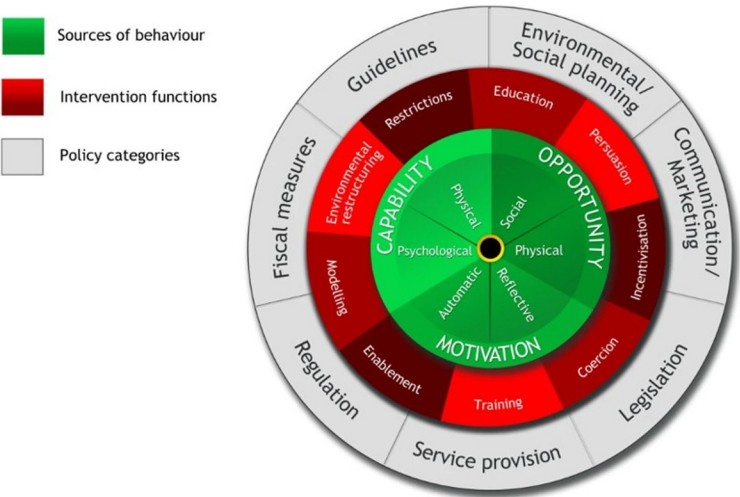

**Fig 1. Behaviour Change Wheel (Michie et al, 2011) [28].**

A systematic review was undertaken to collate evidence to answer the following questions: 1) are behaviour change nutrition- interventions effective in improving maternal and child nutrition in sub-Saharan? 2) which functions of behaviour change interventions are associated with improvements in maternal and child nutrition outcomes?

## Materials and method

### Search strategy and study selection

This systematic review follows the Centre for Reviews and Dissemination [29] and PRISMA guidelines (S1 Checklist) [30, 31]. It is PROSPERO registered (135054). Articles were systematically searched for on major medical and social science databases including Cochrane library, EMBASE, MEDLINE, PSYCHINFO, CINAHL and African Journal OnLine (AJOL) published up to 1st November 2022. The search strategy was based on the most prevalent behaviour

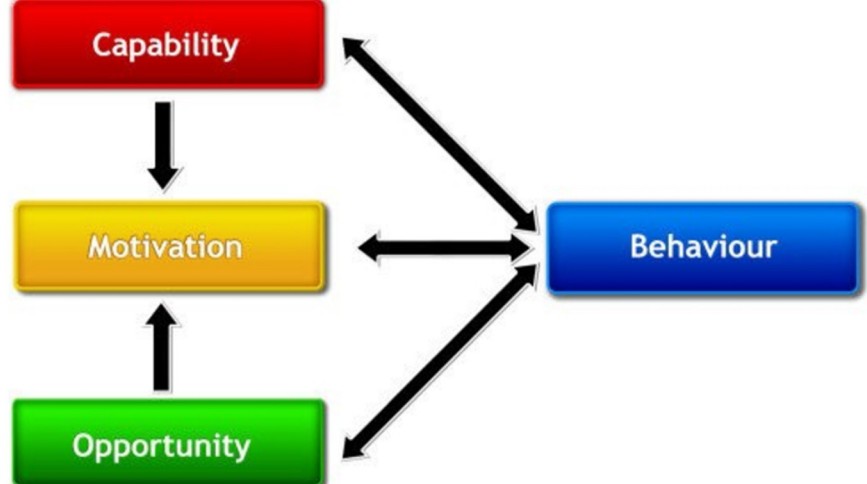

**Fig 2. The COM-B model (capability, motivation, opportunity–behaviour) (Michie et al, 2011) [28].**

change concepts and theories identified by Davis et al (2015) [32]. Nutrition-specific and nutrition-sensitive interventions, and maternal and child nutrition terms were derived from the 2013 Lancet series on Maternal and Child Nutrition [4, 5] and in consultation with experts. The terms were entered into the databases using a combination of Medical Subject Headings (MeSH) and free text to include behaviour change interventions for maternal and child nutrition in sub-Saharan Africa (S1 Table). The terms were based on English and American spelling and were not translated into other languages. Further studies were sought by hand-searching bibliographies of included studies, tracking citations on Google Scholar, and asking experts on the topic. Breastfeeding interventions were excluded as there have been previous systematic reviews on this topic, which incidentally conclude that evidence for including behaviour change in breastfeeding and complimentary feeding interventions is lacking [33, 34].

Publications were stored in Endnote X9.2, duplications were removed and papers were screened on Rayyan qcri online website [35]. Paper titles, abstracts and full text were double-screened. Inclusion criteria can be found in Tables 1 and S2. Full texts were screened by a Health Psychologist to select only interventions that had a behaviour change component and analyse the findings using the Behaviour Change Wheel [28] (S3 Table). Discrepancies were resolved by team review, which was deemed the final consensus.

## Data extraction and quality assessment

Data were extracted by two independent researchers to create Table 3. The table describes all interventions which were categorised by the number of Behaviour Change Intervention

**Table 1. Systematic review inclusion and exclusion criteria.**

|  | Inclusion criteria | Exclusion criteria |
|---|---|---|
| **Publication type** | Journal articles | Protocols<br>Commentary articles<br>Conference papers<br>Abstracts<br>Theses |
| **Study design** | Interventional studies with or without control group<br>Experimental or quasi-experimental design<br>Primary data collection<br>Qualitative evaluations of an intervention | Observational studies<br>Qualitative methodologies to explore perceptions of nutrition<br>Demographic surveillance to investigate nutrition status |
| **Population** | At least one of the following populations living in sub-Saharan Africa:<br>women of child-bearing age (preconception), including adolescents<br>pregnant women<br>fathers<br>infants, children and adolescents | Populations living outside sub-Saharan Africa<br>Expats living in sub-Saharan Africa |
| **Interventions** | Nutrition related behaviour change interventions | Solely breastfeeding or complimentary feeding interventionsNon-behaviour change interventionsNon-nutrition interventions solely focused on<br>HIV/AIDS,<br>malaria<br>other infectious diseases |
| **Outcomes** | At least one of the following outcomes for:<br>Body composition, nutrient intake, dietary patterns<br>Health outcomes including physical and psychological<br>Behavioural or psychosocial outcomes related to nutrition<br>Short- and long-term outcomes |  |

Functions identified, by risk of bias score and publication date. Study designs were coded using an established method [36] based on the functions from the Behaviour Change Wheel and coded to the COM-B model [28]. This is a well-evidenced behaviour change framework and this type of analysis has been carried out in other research studies [28]. Each intervention was reviewed independently by two trainee Health Psychologists, who identified which Behaviour Change Wheel intervention functions were included in each intervention, under the supervision of a chartered Health Psychologist. The results section was structured around the main nutrition, psychosocial and behavioural outcomes, as advised by a nutritionist expert. Studies that included one of these outcomes were described under the appropriate heading.

A quality assessment tool was adapted from a systematic review on digital interventions using mixed methods and tailored for use in this review (S4 and S5 Tables) [37], based on the Centre for Reviews and Dissemination quality assessment criteria [29]. Papers were assessed for risk of bias by two reviewers independently (Table 2). Scores between -4 and 4 indicated a medium risk of bias. Scores below—4 and above 4 indicated a high and low risk of bias respectively.

## Data synthesis

As there were different effect measures used across studies, a meta-analysis was not appropriate. Instead we conducted vote counting based on the direction of effect, as advised by the Synthesis Without Meta-analysis (SWiM) reporting guidelines (S2 Checklist) [38] and the Cochrane handbook for systematic reviews of interventions [39]. Cochrane's most recent advice for accurate vote counting was to consider the direction, not the significance, of each intervention's effect estimates in terms of showing a health benefit or harm, in order to produce a standardised binary metric. To determine if an intervention showed a health benefit or harm, we hypothesised that behaviour change nutrition interventions would improve maternal and child nutrition and psychosocial outcomes in communities in sub-Saharan Africa. A health benefit would include, for example, interventions that reduced infant stunting and wasting, increased the quality and quantity of household foods being eaten and improved mother's and family's psychological capabilities to feed their children. Each outcome from each paper was classified as either positive (supports the hypothesis–health benefit), negative (rejects the hypothesis–health harm) or as inconclusive if outcome could not be determined. The vote-counting results were summarized visually using an effect direction plot. Arrows were used to represent the combined direction for each study and outcome in the effect direction plot. Based on previous criteria [39], more than 70% of positive outcomes were interpreted as overall positive (↑), and studies with more than 70% of negative outcomes as overall negative (↓), whereas less than 70% of either positive or negative outcomes within one outcome category were interpreted as inconclusive (↔). Studies were excluded from vote counting if they did not demonstrate statistical evidence to answer the hypothesis including process evaluations and qualitative studies, and also studies graded as high risk of bias. Within studies, outcomes that could be interpreted ambiguously as to whether this was a benefit or harm within the context, for example increased use of oil, were also excluded. Each outcome was categorised into one of four overarching categories: 1) anthropometric markers, 2) dietary outcomes, 3) psychosocial outcomes, 4) other outcomes. Anthropometric markers included measure of infant body composition such as weight-age-Z scores (HAZ), weight-height-Z scores (WHZ), weight-age-Z scores (WAZ). Dietary outcomes included potential changes related to a variety of individual and household dietary diversity measures and food items consumed. Psychosocial outcomes included potential changes in nutrition knowledge, practice, self-efficacy and developmental outcomes. Other outcomes encompassed outcomes that indirectly influence

**Table 2. Risk of Bias score table.**

| # | Author, year | Study design | Randomisation method | Blinding | Difference between groups | Selection | Loss to follow-up | Dietary assessment | Behaviour change component | Performance bias | Intention to Treat | Analysis | Confounding | Total | Risk of Bias |
|---|---|---|---|---|---|---|---|---|---|---|---|---|---|---|---|
| 1 | Abiyu, 2020 | +1 | +1 | 0 | +1 | +1 | +1 | 0 | 0 | 0 | +1 | +1 | +1 | 8 | Low |
| 2 | Antwi, 2020 | +1 | +1 | -1 | 0 | 0 | +1 | 0 | +1 | 0 | -1 | +1 | -1 | 2 | Medium |
| 3 | Aubel, 2004 | 0 | -1 | -1 | -1 | 0 | -1 | 0 | 0 | 0 | N/A | 0 | -1 | 0 | High |
| 4 | Becquey, 2019 | +1 | +1 | -1 | -1 | +1 | 0 | 0 | 0 | 0 | +1 | +1 | +1 | 4 | Medium |
| 5 | Becquey, 2022 | +1 | +1 | -1 | +1 | +1 | 0 | 0 | +1 | -1 | +1 | +1 | +1 | 6 | Low |
| 6 | Briaux, 2020 | +1 | +1 | -1 | +1 | +1 | 0 | 0 | +1 | -1 | +1 | +1 | +1 | 4 | Medium |
| 7 | Byrd, 2019 | +1 | +1 | 0 | -1 | -1 | 0 | 0 | -1 | +1 | -1 | -1 | -1 | 7 | Low |
| 8 | DeLorme, 2018 | -1 | -1 | N/A | -1 | -1 | +1 | 0 | 0 | 0 | N/A | +1 | -1 | -3 | Medium |
| 9 | Demilew, 2020 | +1 | +1 | 0 | +1 | +1 | +1 | +1 | +1 | 0 | 0 | +1 | +1 | 9 | Low |
| 10 | Diddana, 2018 | +1 | +1 | 0 | +1 | +1 | 0 | -1 | +1 | 0 | N/A | +1 | +1 | 6 | Low |
| 11 | Downs, 2019 | -1 | -1 | -1 | N/A | 0 | +1 | 0 | +1 | +1 | N/A | 0 | 0 | 0 | Medium |
| 12 | Dozio, 2016 | -1 | -1 | -1 | -1 | 0 | -1 | 0 | +1 | 0 | N/A | -1 | -1 | -6 | High |
| 13 | Ezezika, 2018 | -1 | -1 | N/A | 0 | -1 | +1 | 0 | -1 | 0 | N/A | -1 | N/A | -4 | High |
| 14 | Felegush, 2018 | +1 | +1 | -1 | +1 | +1 | +1 | +1 | +1 | 0 | N/A | +1 | +1 | 8 | Low |
| 15 | Fernandes, 2016 | -1 | +1 | -1 | 0 | +1 | -1 | -1 | +1 | -1 | N/A | -1 | -1 | -4 | High |
| 16 | Flax, 2021 | +1 | +1 | -1 | 0 | +1 | +1 | 0 | 0 | 0 | -1 | +1 | +1 | 4 | Medium |
| 17 | Flax, 2022 | 0 | 0 | -1 | -1 | +1 | N/A | 0 | +1 | 0 | N/A | +1 | +1 | 2 | Medium |
| 18 | Galasso, 2009 | 0 | +1 | -1 | 0 | +1 | -1 | +1 | 0 | 0 | +1 | +1 | +1 | 4 | Medium |
| 19 | Galasso, 2019 | +1 | +1 | -1 | +1 | +1 | +1 | +1 | 0 | +1 | +1 | +1 | +1 | 9 | Low |
| 20 | Gelli, 2018 | +1 | +1 | +1 | +1 | +1 | +1 | +1 | +1 | +1 | +1 | +1 | -1 | 10 | Low |
| 21 | Gelli, 2020 | +1 | 0 | +1 | +1 | +1 | +1 | +1 | +1 | +1 | +1 | +1 | -1 | 10 | Low |
| 22 | Grant, 2022 | 0 | -1 | -1 | N/A | +1 | N/A | 0 | 0 | 0 | N/A | +1 | +1 | 1 | Medium |
| 23 | Han, 2021 | +1 | +1 | -1 | +1 | +1 | 0 | +1 | +1 | 0 | +1 | 0 | +1 | 7 | Low |
| 24 | Hurley | 0 | -1 | -1 | 0 | 0 | +1 | +1 | 0 | 0 | N/A | +1 | +1 | 2 | Medium |
| 25 | Huybregts, 2019 | +1 | +1 | 0 | 0 | +1 | 0 | -1 | +1 | 0 | +1 | +1 | +1 | 6 | Low |
| 26 | Jacobs, 2013 | +1 | +1 | -1 | 0 | +1 | -1 | -1 | +1 | 0 | N/A | +1 | -1 | 1 | Medium |
| 27 | Jemmott, 2011; | +1 | +1 | +1 | +1 | +1 | +1 | 0 | +1 | +1 | +1 | +1 | +1 | 11 | Low |
| 28 | Jemmott, 2019 | +1 | +1 | +1 | +1 | 0 | +1 | 0 | +1 | +1 | +1 | +1 | +1 | 10 | Low |
| 29 | Kang, 2017a | +1 | +1 | -1 | -1 | -1 | +1 | -1 | -1 | +1 | N/A | +1 | -1 | -1 | Medium |
| 30 | Kang, 2017b | 0 | -1 | -1 | -1 | 0 | -1 | +1 | -1 | +1 | 0 | 0 | -1 | -4 | Medium |
| 31 | Kang, 2017c | +1 | +1 | -1 | -1 | -1 | +1 | -1 | -1 | +1 | N/A | +1 | -1 | -1 | Medium |
| 32 | Katenga-Kaunda 2020 | +1 | +1 | -1 | 0 | 0 | +1 | 0 | +1 | -1 | -1 | +1 | +1 | 3 | Medium |
| 33 | Katenga-Kaunda 2021 | +1 | +1 | -1 | +1 | +1 | 0 | 0 | 0 | 0 | -1 | +1 | +1 | 4 | Medium |
| 34 | Katenga-Kaunda, 2022 | 0 | +1 | N/A | N/A | 0 | N/A | 0 | +1 | -1 | N/A | +1 | +1 | 3 | Medium |
| 35 | Kim, 2015 | -1 | -1 | -1 | -1 | -1 | 1 | -1 | -1 | 0 | N/A | 0 | -1 | -7 | High |
| 36 | Kim, 2016 | 0 | 1 | -1 | 1 | 1 | -1 | 1 | 0 | 0 | -1 | 1 | 1 | 3 | Medium |
| 37 | Kim, 2019a | +1 | +1 | 0 | 0 | +1 | +1 | 0 | +1 | 0 | +1 | +1 | +1 | 8 | Low |
| 38 | Kim, 2019b | 0 | -1 | -1 | -1 | 0 | N/A | 0 | 0 | 0 | N/A | +1 | +1 | -1 | Medium |
| 39 | Kumar, 2018 | +1 | +1 | -1 | +1 | 0 | -1 | 0 | +1 | 0 | +1 | +1 | +1 | 5 | Low |
| 40 | Rosenberg, 2018 | +1 | +1 | -1 | +1 | 0 | -1 | 0 | +1 | 0 | +1 | +1 | +1 | 5 | Low |
| 41 | Lagerkvist, 2018 | -1 | +1 | -1 | +1 | +1 | -1 | -1 | +1 | 0 | N/A | +1 | +1 | 2 | Medium |
| 42 | Leroy, 2016 | +1 | +1 | -1 | +1 | 0 | 0 | 0 | 0 | +1 | +1 | +1 | +1 | 6 | Low |
| 43 | Leroy, 2018 | +1 | +1 | -1 | +1 | 0 | 0 | 0 | 0 | +1 | +1 | +1 | +1 | 6 | Low |
| 44 | Leroy, 2019 | +1 | +1 | -1 | +1 | 0 | 0 | 0 | 0 | +1 | +1 | +1 | +1 | 6 | Low |
| 45 | Olney, 2019 | +1 | +1 | -1 | +1 | 0 | 0 | 0 | 0 | +1 | +1 | +1 | +1 | 6 | Low |
| 46 | Heckert, 2020 | 0 | -1 | +1 | +1 | 0 | N/A | N/A | 0 | +1 | N/A | +1 | +1 | 4 | Medium |

*(Continued)*

**Table 2.** (Continued)

| # | Author, year | Study design | Randomisation method | Blinding | Difference between groups | Selection | Loss to follow-up | Dietary assessment | Behaviour change component | Performance bias | Intention to Treat | Analysis | Confounding | Total | Risk of Bias |
|---|---|---|---|---|---|---|---|---|---|---|---|---|---|---|---|
| 47 | Lion, 2018 | 0 | +1 | -1 | 0 | +1 | 0 | -1 | +1 | +1 | N/A | +1 | +1 | **4** | Medium |
| 48 | McKune, 2020 | +1 | +1 | +1 | +1 | +1 | -1 | +1 | 0 | +1 | -1 | +1 | +1 | **7** | Medium |
| 49 | Mlinda, 2018 | +1 | +1 | -1 | +1 | +1 | +1 | 0 | 0 | 0 | +1 | +1 | +1 | **7** | Low |
| 50 | Morris, 2012 | -1 | -1 | 0 | +1 | -1 | +1 | 0 | +1 | 0 | +1 | +1 | +1 | **3** | Medium |
| 51 | Muehlhoff, 2016 | 0 | -1 | -1 | -1 | 0 | +1 | -1 | -1 | -1 | N/A | -1 | -1 | **-7** | High |
| 52 | Mushaphi, 2015 | 0 | +1 | -1 | +1 | -1 | -1 | +1 | -1 | 0 | N/A | 0 | -1 | **-2** | Medium |
| 53 | Mushaphi, 2017 | 0 | +1 | -1 | +1 | -1 | -1 | +1 | -1 | 0 | N/A | 0 | -1 | **-2** | Medium |
| 54 | Mutiso, 2018 | -1 | +1 | -1 | -1 | +1 | -1 | 0 | +1 | 0 | N/A | +1 | +1 | **1** | Medium |
| 55 | Nordhagen, 2018 | -1 | +1 | -1 | N/A | +1 | N/A | 0 | 0 | -1 | N/A | -1 | -1 | **-3** | Medium |
| 56 | Ogunsile & Ogundele, 2016 | 0 | +1 | -1 | 0 | +1 | +1 | 0 | +1 | +1 | n/a | +1 | -1 | **4** | Medium |
| 57 | Olney, 2015 | +1 | +1 | 0 | +1 | +1 | 0 | +1 | +1 | +1 | N/A | +1 | +1 | **9** | Low |
| 58 | Olney, 2016 | +1 | +1 | 0 | +1 | +1 | 0 | +1 | +1 | +1 | N/A | +1 | +1 | **9** | Low |
| 59 | Nielsen, 2018 | 0 | +1 | 0 | +1 | +1 | 0 | 0 | 0 | +1 | N/A | +1 | +1 | **6** | Low |
| 60 | Passarelli, 2020 | +1 | +1 | -1 | 0 | +1 | +1 | 0 | +1 | +1 | -1 | +1 | +1 | **6** | Low |
| 61 | Ruel-Bergeron, 2019a | -1 | -1 | -1 | -1 | +1 | -1 | +1 | 0 | -1 | -1 | -1 | -1 | **-6** | High |
| 62 | Ruel-Bergeron, 2019b | 0 | -1 | -1 | -1 | +1 | -1 | N/A | 0 | 0 | N/A | 0 | N/A | **-3** | Medium |
| 63 | Christian, 2020 | 0 | -1 | -1 | -1 | +1 | -1 | 0 | 0 | -1 | -1 | -1 | -1 | **-7** | High |
| 64 | Saaka, 2011 | 0 | -1 | -1 | -1 | 0 | -1 | -1 | -1 | 0 | N/A | -1 | -1 | **-7** | High |
| 65 | Saaka, 2021 | 0 | +1 | -1 | 0 | +1 | -1 | +1 | 0 | 0 | N/A | +1 | +1 | **3** | Medium |
| 66 | Skar, 2019 | 0 | 0 | -1 | 0 | 0 | -1 | +1 | -1 | 0 | N/A | +1 | -1 | **-2** | Medium |
| 67 | Santoso, 2021 | +1 | +1 | 0 | +1 | -1 | +1 | +1 | -1 | -1 | +1 | +1 | +1 | **5** | Low |
| 68 | Shapu, 2022 | +1 | +1 | 0 | +1 | +1 | +1 | 0 | +1 | +1 | +1 | 0 | +1 | **9** | Low |
| 69 | Steyn, 2015 | +1 | +1 | 0 | 0 | +1 | +1 | 0 | -1 | 0 | N/A | 0 | -1 | **2** | Medium |
| 70 | De Villiers, 2016 | +1 | +1 | 0 | 0 | +1 | +1 | 0 | -1 | 0 | N/A | 0 | -1 | **2** | Medium |
| 71 | Tamiru, 2017 | 0 | +1 | -1 | +1 | 0 | -1 | +1 | +1 | +1 | n/a | 0 | +1 | **4** | Medium |
| 72 | Tariku, 2015; | +1 | +1 | -1 | +1 | +1 | +1 | 0 | +1 | 0 | n/a | +1 | 0 | **6** | Low |
| 73 | Mulualem, 2016 | +1 | +1 | -1 | +1 | +1 | +1 | 0 | +1 | 0 | n/a | +1 | 0 | **6** | Low |
| 74 | Thuita, 2015; | 0 | -1 | -1 | -1 | 0 | -1 | -1 | -1 | -1 | N/A | -1 | -1 | **-9** | High |
| 75 | Mukuria, 2016 | 0 | -1 | -1 | -1 | 0 | -1 | -1 | -1 | -1 | N/A | -1 | -1 | **-9** | High |
| 76 | Tomlinson, 2020 | 0 | +1 | -1 | -1 | +1 | 0 | +1 | +1 | 0 | -1 | +1 | -1 | **1** | Medium |
| 77 | Walton, 2017 | +1 | +1 | +1 | +1 | +1 | +1 | 0 | +1 | -1 | N/A | +1 | +1 | **8** | Low |
| 78 | Workicho, 2021 | 0 | -1 | -1 | +1 | -1 | +1 | 0 | 0 | 0 | N/A | 0 | +1 | **-1** | Medium |
| 79 | Yetnayet, 2017 | 0 | +1 | -1 | +1 | 0 | -1 | 0 | +1 | 0 | N/A | +1 | +1 | **3** | Medium |

nutrition such as prevalence of infectious disease, behaviours related to hygiene and economic and agricultural indicators.

Results are presented as a narrative synthesis [40] and based on the intervention's quality assessment and direction of effect in relation to research question one. Each finding from a study reports the risk of bias score (RoB) and direction of effect (DE) in brackets to accompany

**Table 3. Descriptions of included studies found in the systematic review.**

| | Author, publication year, country | Study design and intervention | Behaviour change wheel intervention function | Setting and participants | Assessment | Outcome and Take-home message | Risk of Bias Score |
|---|---|---|---|---|---|---|---|
| | | | | **1 BCW IF** | | | |
| 1 | Shapu et al, 2022, Nigeria | Cluster randomized controlled trial of the triple benefit health education guided by the information, motivation, behavioural skill (IMB) theory in Nigeria. | Education. | Adolescent girls aged 10 to 19 years old in Maiduguri, Borno state, Nigeria. | **Nutrition** Food security questions. **Psychosocial** Knowledge and attitude questionnaires were adapted from the Food and Agricultural Organisation of the United Nations. | Significant difference in knowledge, attitude and food security at three and six-months post intervention. The triple benefit health education intervention package employed in this study can serve as an intervention tool to combat malnutrition among adolescent girls in Nigeria at large. | Low 9 |
| 2 | Diddana et al, 2018, Ethiopia | A community-based cluster RCT nutrition education programme based on Health Belief Model and nutritional knowledge and dietary practice of pregnant women. | Education. | 138 pregnant women who were permanent residents in Dissie, Ethiopia. | **Nutrition** Nutritional knowledge was collected by using 15 nutrition knowledge questions. **Psychosocial** Stage of pregnancy, Health Belief Model constructs, nutrition knowledge, and dietary practices collected using structured questionnaires. | Nutritional knowledge significantly increased in the intervention group over time, but was no significant difference between intervention and control. There was significant improvement in the scores of Health Belief Model constructs in intervention group. Providing nutrition education based on Health Belief Model improves nutritional knowledge and dietary practices of pregnant women. | Low 6 |
| 3 | Downs et al, 2019, Senegal | Before and after surveys conducted as part of a pilot study examining the implementation and impact of a mHealth voice messaging intervention targeting IYCF practices | Persuasion | Mothers and fathers in all households in the villages' farming group with a child 6 to 24 months in three rural villages in Senegal (n=47) | 24-h recall (MDD, MMF, and MAD indicators), FFQ Beliefs, attitudes, and intentions survey included sociodemographic information (age, wealth, education, and literacy), questions related to IYCF beliefs, attitudes, intentions, and behaviours as well exposure to the intervention | Three of the eight behaviours increased and one decreased Significant increase in the number of children that consumed fish Significantly higher frequency of egg, fish, and thick porridge consumption Findings suggest that voice messaging IYCF interventions in Senegal have the potential to improve IYCF behaviours among young children in the short term | Medium 0 |
| 4 | Kang et al, 2017a, Ethiopia | Impact evaluation of a community-based participatory nutrition promotion programme, involving a 2-week group nutrition session, attempted to improve child feeding and hygiene. | Enablement | Participants were 2064 mother–child pairs from Habro and Melka Bello districts, Ethiopia | A total of thirteen items in the observation checklist were conducted during the programme (n=114). A recall survey was conducted to explore the participants' perception of the nutrition sessions. | Mothers in the intervention area showed higher scores than those in the control area regarding meal frequency and feeding score. There were no differences in the scores of breast-feeding, dietary diversity and hand washing between the two areas. The results can be applied to scaling up programme and designing similar social and behaviour change communication models by governments and non-governmental organisations in Ethiopia or other East African communities. | Medium -1 |
| 5 | Kang et al 2017c, Ethiopia | Impact evaluation of a community-based participatory nutrition promotion programme, involving a 2-week group nutrition session, attempted to improve child feeding and hygiene. | Enablement | Participants were 2064 mother–child pairs from Habro and Melka Bello districts, Ethiopia. | A total of thirteen items in the observation checklist were conducted during the programme (n=114). A recall survey was conducted to explore the participants' perception of the nutrition sessions. | Compared with children 6 to 24 months of age in the control area, those in the intervention area had a greater increase in z scores for length-for-age and weight-for-length At the end of the 12-month follow-up, children in the intervention area showed an 8.1% (P = 0.02) and 6.3% (P = 0.046) lower prevalence of stunting and underweight, respectively | Medium -1 |
| 6 | Kang et al, 2017b, Ethiopia | Process evaluation of a community-based participatory nutrition (CBPN) promotion programme, involving a 2-week group nutrition session, attempted to improve child feeding and hygiene. | Enablement | 1987 mothers from Habro and Melka Bello districts, Ethiopia. | **Nutrition** Records of 372 nutrition sessions, as part of a cluster-randomized trial, among mothers from a household survey and CPNP participants (n 197) from a recall survey. | The recall survey among participants showed a positive perception of the sessions (~90%) and a moderate level of message recall (~65%). The household survey found that the CPNP participants had higher minimum dietary diversity at the early stage and a higher involvement in the Essential Nutrition Action (ENA) programme over a year of follow-up compared with non-participants within the intervention area. | Medium -4 |

*(Continued)*

**Table 3.** (Continued)

| | Author, publication year, country | Study design and intervention | Behaviour change wheel intervention function | Setting and participants | Assessment | Outcome and Take-home message | Risk of Bias Score |
|---|---|---|---|---|---|---|---|
| | | | | **2 BCW IF** | | | |
| 7 | Demilew et al, 2020, Ethiopia | A two-arm parallel cluster RCT on the effect of guided counselling using the Health Belief Model and the Theory of Planned Behaviour on the nutritional status of pregnant women. | Enablement, Persuasion. | 694 pregnant women from West Gojjam Zone, Ethiopia. | **Nutrition** MUAC measurements. Food security status was assessed using 27 previously validated questions. **Psychosocial** Structured questionnaires through one-to-one interview on socio-demographic variables, obstetric history, Health Belief Model, and Theory Planned Belief constructs. **Other** The Wealth Index | After the intervention, the prevalence of undernutrition was 16.7% lower in the intervention group compared with the control arm. Women in the intervention group showed significant improvements in nutritional status at the end of the trial than the control group. | Low 9 |
| 8 | Felegush et al, 2018, Ethiopia | RCT peer-led nutrition education intervention promoting locally available pulses. | Environmental restructuring, Training. | 202 school children in Ethiopia. | **Nutrition** Pre-test, post-test and HAZ, BMI for age, DDS (n=202) **Psychosocial** Knowledge, attitude and practice (KAP) questionnaire with 23 items was administered (n=202). | Mean DDS significantly increased. Increased knowledge, attitude and practice about pulse preparation and consumption. There was no significant difference in nutritional status but there was decreased prevalence of wasting. Peer led education strategy provides an opportunity to reduce malnutrition and its impacts if properly designed, including the use of behavioural change mode. | Low 8 |
| 9 | Becquey et al, 2019, Burkina Faso | Cluster-RCT integration of a preventive package including age-appropriate behaviour change communication on nutrition, health, and hygiene practices and a monthly supply of small-quantity lipid-based nutrient supplements. | Enablement, Persuasion. | Children 0–59 months and uses both community and health facility platforms in in Burkina Faso's Northern Region. | **Nutrition** MUAC, WLZ. **Other** Household wealth status score. | There were no impacts on either acute malnutrition treatment coverage or incidence. This is discussed as barriers to uptake of preventive and treatment services at the health centre. | Medium 4 |
| 10 | Tamiru et al 2017, Ethiopia | Quasi experimental evaluation of the effectiveness of school-based nutrition education using peer-led, health promotion through school media and health clubs. | Education. Modelling. | Four primary schools (2 rural, 2 urban) in Jimma Zone, Oromia Regional State, Ethiopia. | Focus groups with children and interviews with school principals and relevant teachers. Observations conducted of school environment and review of the nutrition policy. | Significant improvements in food variety between food secure and insecure households, and improvements in animal dietary intake due to intervention. School based nutrition education programme should be implemented in comprehensive school health programme to reach students and their families. | Medium 4 |
| 11 | Morris et al, 2012, Uganda | National wide impact of combining a group-based psychosocial intervention with an existing emergency feeding programme for internally displaced mothers. | Education, Enablement. | 237 displaced mothers from Kitgum district in Northern Uganda. | **Psychosocial** The Acholi Home Observation for Measurement of the Environment (50 items) was used to assess the impact of the intervention on mother's ability to stimulate her child. The Kitgum maternal mood scale to assess changes in maternal mood. The Knowledge, Attitudes and Practice (KAP) questionnaire (10 items). | The interventions showed that mothers had greater involvement with their babies, more availability of play materials, and less sadness and worry. A proportion of the mothers chose to continue the intervention spontaneously with other mothers in their neighbourhoods. Further research needs to explore the longer-term impact on child growth and intellectual development as well as maternal mood. | Medium 3 |

*(Continued)*

**Table 3.** (Continued)

| | Author, publication year, country | Study design and intervention | Behaviour change wheel intervention function | Setting and participants | Assessment | Outcome and Take-home message | Risk of Bias Score |
|---|---|---|---|---|---|---|---|
| 12 | Antwi et al, 2020, Ghana | Pre-post controlled design was used to evaluate the programme to assess the effect of nutrition education that is incorporated into the existing physical education (PE) lessons. | Education, Training | 351 school-going children aged 6–12, and their primary home-based caregivers, and PE teachers in the study intervention Accra. | **Psychosocial** Nutrition knowledge, attitude and practices (KAP) scores **Nutrition** Single 24-hour recall and food frequency methods. (HAZ), (WAZ) (WHZ) and BMI-for-age Z scores. | Intervention groups had significantly higher nutrition knowledge scores compared to controls in the lower primary level. A higher proportion of children in the intervention group strongly agreed they enjoyed learning about food and nutrition issues compared to the control group. There was no significant difference in MDD scores or in measured anthropometric indices, with a marginal reduction in stunting in the intervention group. Nutrition knowledge of teachers and caregivers significantly improved. | Medium 2 |
| 13 | Mushaphi et al, 2017, South Africa | Pretest–posttest control group design. Nutrition Education Intervention Programmeme comprised ten topics emphasising healthy eating, hygiene and sanitation. | Education. Enablement. | Eight villages, Limpopo, South Africa. A total of 129 children aged 3 -5 years and 125 caregivers were included. | Interviews with caregivers, on feeding practices and nutrition knowledge. **Nutrition** Anthropometric measurements and blood samples (iron and vitamin A status) of children were taken. | Majority of children in both the experimental and control groups were given three meals or more per day, including starchy and protein rich foods at baseline and post intervention. Intake of mixed traditional dishes and some indigenous foods did improve significantly. The intake of mixed traditional dishes as well as the intake of the indigenous foods, stinging nettle, meldar, wild peach, pineapple, dovhi, tshigume and thophi, increased significantly in both the experimental and control groups. | Medium -2 |
| 14 | Mushaphi et al, 2015, South Africa | Pretest–posttest control group design. Nutrition Education Intervention Programme comprised ten topics emphasising healthy eating, hygiene and sanitation. | Education. Enablement. | Eight villages, Limpopo, South Africa. A total of 129 children aged 3 -5 years and 125 caregivers were included. | Interviews with caregivers, on feeding practices and nutrition knowledge. **Nutrition** Anthropometric measurements and blood samples (iron and vitamin A status) of children were taken. | The nutritional status of children did not change significantly following intervention. According to the categories for indicators of iron status, the number of children who were in the 'adequate' category for serum iron, serum ferritin, serum transferrin and percentage transferrin saturation did not change in both groups at postintervention assessment Authors stated that no significant effect on nutrition status was possibly due to the interventions' short duration (12 months) and that food supplementation was not included. | Medium -2 |
| 15 | DeLorme et al, 2018, Kenya | Analysis of focus group evaluation data of the Kanayakla nutrition programme to engage social networks to improve infant and young child feeding. | Enablement, Training. | Parents with infants and young children were recruited from rural villages in Kenya's East region as well as other community members (neighbours, grandparents, older parents). | Focus group data (four groups; n = 35, 7 men and 28 women) | The intervention increased nutrition knowledge and confidence, changed perceptions, and supported infant and child feeding practices at the individual, interpersonal, and institutional levels. Environmental and economic constraints continue to affect food access. Engaging the household's network of interpersonal and community relationships can play a role in addressing structural barriers to improve nutrition. | High -3 |
| 16 | Dozio et al, 2016, Central Africa Republic | Evaluation of non-government organisation intervention that implemented food coupons and therapy groups to pregnant and lactating women. | Incentivisation, Education. | 900 pregnant and lactating women received food coupons and 199 women who were identified as most psychologically vulnerable received therapy in Central African Republic. | Interviews with 86 pregnant and lactating women. **Nutrition** IDDS, FCS **Psychosocial** Knowledge, attitude and practice (KAP) questionnaire with 23 items was administered. | Women's average IDDS significantly increased, households improved their FCS, and psychological wellbeing improved. Multi-sectoral approach strengthens family resilience, and psychological influences should be considered to improve nutrition. | High -6 |
| 17 | Saaka et al, 2011, Ghana | A pre-test/post-test Evaluation (surveys) design of the Catholic Relief Services' | Education, Training. | The programme included women and children living in food insecure households in | A pre-test/post-test design involving two cross sectional surveys at baseline and follow-up surveys were used to | The programme reduced chronic malnutrition by 1.5 percentage points per year, and empowered individuals and communities to adopt positive health behaviours. | High -7 |

*(Continued)*

**Table 3.** (*Continued*)

| | Author, publication year, country | Study design and intervention | Behaviour change wheel intervention function | Setting and participants | Assessment | Outcome and Take-home message | Risk of Bias Score |
|---|---|---|---|---|---|---|---|
| | | Development Assistance Programme which promoted infant feeding practices, community health seeking behaviour, food supplies and support groups. | rural farming communities. Communities were located in select districts in three northern regions of Ghana covered 221 beneficiary communities. Evaluation included 30 cluster community and 16 interviews were conducted in each cluster (n=480) 10 baseline survey interviews per cluster (n=300). | determine the outcome/impact of the programme. Interviews were conducted using pre-designed questionnaires.<br><br>**Nutrition**<br>Pre-school children's weight and height were measured. | The programme showed improvement over time for health-seeking behaviours, practices and coverage of health and nutrition services. Community involvement and ownership were central to the continuity of health activities, where individuals championed and promoted the programme. | | |
| | | | | **3 BCW IF** | | | |
| 18 | Jemmott, 2011, South Africa | Cluster RCT Cognitive-behavioural health-promotion intervention. Interventions were based on Social Cognitive Theory, and the Theory of Planned Behaviour. | Enablement, Training, Education. | Participants were 1057 grade 6 learners (mean age =12.4 years), with 96.7% retained at 12-month follow-up, from Eastern Cape Province, South Africa. | **Nutrition**<br>A seven-item food frequency questionnaire to evaluate 5-a-Day studies was used to measure fruit and vegetable consumption over the past 30 days (n=1056).<br><br>**Psychosocial**<br>Health promoting-behaviour attitude and intention and drug-and-alcohol-use attitude and intention were assessed on 5-point rating scales (n=1056). | Participants in the health-promotion intervention met 5-a-day fruit and vegetable and physical activity guidelines.<br>The intervention increased health-promotion knowledge, attitude and intention.<br>Theory based and contextually appropriate interventions may increase health behaviours among young adolescents in sub-Saharan Africa. | Low 11 |
| 19 | Jemmott, 2019, South Africa | Cluster RCT Cognitive-behavioural health-promotion intervention. Interventions were based on Social Cognitive Theory, and the Theory of Planned Behaviour. | Enablement, Training, Education. | Participants were 1057 grade 6 learners (mean age =12.4 years), with 96.7% retained at 12-month follow-up, from Eastern Cape Province, South Africa. | **Nutrition**<br>A seven-item food frequency questionnaire to evaluate 5-a-Day studies was used to measure fruit and vegetable consumption over the past 30 days (n=1056).<br><br>**Psychosocial**<br>Health promoting-behaviour attitude and intention and drug-and-alcohol-use attitude and intention were assessed on 5-point rating scales (n=1056). | The effect on 5-a-day diet did not weaken at long-term compared with short-term follow-up. The effect on physical activity guidelines was weaker at long-term follow-up, mainly because of a reduced effect on muscle-strengthening physical activity.<br>The intervention also increased health promotion attitude and intention and health knowledge and reduced binge drinking compared with the control group. | Low 10 |
| 20 | Galasso et al, 2019, Madagascar | Large scale community-based RCT nutrition programme that included weighing infants, providing nutrition counselling to the mothers, promoting behavioural change nutrition and hygiene education sessions, and suppling micronutrient supplementation. | Training, Education, Enablement. | 3738 mothers and infants in 1999 in four provinces and expanded to all six provinces in Madagascar in 2000. | **Nutrition**<br>Child LAZ and WLZ.<br>Five tablet-based questionnaires administered at the three timepoints on household demographics, mother and primary caregiver questionnaire, child questionnaire, community health worker questionnaire and village details. | There were no main effects of any of the intervention groups on any measure of anthropometry or any of the child development outcomes in the full sample<br>Compared with children in the three intervention group, the youngest children (<6 months at baseline) in the T2 and T3 intervention groups who were fully exposed to the child LNS dose had higher length-for-age Z scores and lower stunting prevalence. | Low 9 |

(*Continued*)

**Table 3.** (Continued)

| | Author, publication year, country | Study design and intervention | Behaviour change wheel intervention function | Setting and participants | Assessment | Outcome and Take-home message | Risk of Bias Score |
|---|---|---|---|---|---|---|---|
| 21 | Abiyu et al, 2020, Ethiopia | Cluster-RCT aimed to evaluate the effectiveness of complementary feeding behaviour change communication intervention delivered through community-level actors on the dietary adequacy of infants. | Training, Modelling, Enablement | 612 mothers of infants aged <6 months at enrolment. Rural communities of West Gojjam Zone, Ethiopia. | **Nutrition** A pre-tested, structured interviewer-administered questionnaire MDD, MMF and MAD. 24 hours dietary recall of mothers. | The intervention significantly increased consumption of dairy products, eggs, vitamin A-rich fruits and vegetables, other fruits and vegetables and animal-source foods. Complementary feeding behaviour change communication intervention delivered through community level actors significantly improved the dietary adequacy of infants. | Low 8 |
| 22 | Walton et al, 2017, Kenya | Cross sectional study design nutrition education intervention was developed to enhance women's nutrition knowledge and food skills. | Modelling, Education, Persuasion. | Study was located, Central Province, Kenya. The Mukurweini study group consisted of 88 women in four dairy membership-duration categories from a previous study and non-dairy member women. The intervention (n=55) and control (n=56). | Data was collected using face-to-face interviews, which were pre-intervention survey on household socio-demographic and food security data. **Nutrition** Pre- and post-intervention, women's food intake data were collected using a single four-pass 24-hour diet recall. Nutrition knowledge and practices data was collected using an adapted structured questionnaire. | The approach for women's nutrition education positively influenced nutrition knowledge, practices and diet quality. Increased dietary diversity was dependent on dairy group membership. Some intervention effects were dependent on poverty reduction, but all women were able to make positive dietary changes when informed. Need to examine long term impacts of nutrition education interventions. | Low 8 |
| 23 | Passarelli et al, 2020, Ethiopia | A cluster RCT on chicken production intervention with nutrition-sensitive behaviour change communication. | Environmental restructuring, Enablement, Persuasion. | 829 children aged 0–36 months at baseline from rural agricultural villages in 4 regions of Ethiopia: Amhara; Oromia; Southern Nations, Nationalities, and Peoples' Region; and Tigray. | **Nutrition** Anthropometric measurements: HAZ, WAZ, WHZ at 9 (midline) and 18 months. MDD for children. **Psychosocial** Women's input into decision making related to chickens adapted from the Women's Empowerment in Agriculture Index. **Other** Questionnaire concerning chicken production and household nutrition and health was administered to the woman. | The chicken production intervention without behaviour change communication had higher HAZ and WAZ at end line than the controls. The chicken intervention with behaviour change communication showed similar trends at end line but were not significantly different to the control, but children had higher MDD and egg consumption. A chicken production intervention with or without nutrition-sensitive behaviour change communication may have benefited child nutrition and did not increase morbidity. | Low 6 |
| 24 | Leroy et al, 2019, Burundi | 4-arm cluster-RCT repeated cross-sectional design including food rations, health services strengthening and promotion, and behaviour change communication on nutrition, hygiene, and health practices. | Enablement, Education, Persuasion. | Pregnant women (>4 months of gestation) and mothers of children aged 6-24 months from Burundi. | **Nutrition:** Three repeated cross-sectional surveys were conducted including household dietary diversity, hunger, food insecurity, and food consumption. Also mother diet diversity and child index of infant and young children feeding (6946 households). HHS, HFIAS, MDD. | The intervention significantly improved the percentage of food secure households, increased household energy consumption and micronutrient consumption. Positive effect on maternal dietary diversity (+0.4 food groups, P < 0.05) and increased. The effects on many outcomes were attributable to the food rations Post-programme effects (P < 0.05) were found on household food security, maternal dietary diversity, and younger sibling's complementary feeding practices. | Low 6 |
| 25 | Huybregts et al 2019, Mali | A two-arm cluster-RCT integrating small-quantity lipid-based nutrient supplements into community-level screening for acute malnutrition. | Enablement, Persuasion, Incentivisation. | Children 6–23 months of age and caregivers. 48 health center catchment areas in Mali. | **Nutrition** Anthropometric measurements: WLZ, MUAC. | The intervention significantly increased acute malnutrition screening coverage. No impact on treatment coverage or AM prevalence was found. Children in the intervention arm, however, were 29% less likely to develop a first AM episode (incidence) and, compared to children in comparison arm, their overall risk of AM (longitudinal prevalence) was 30%. | Low 6 |

(*Continued*)

**Table 3.** (Continued)

| | Author, publication year, country | Study design and intervention | Behaviour change wheel intervention function | Setting and participants | Assessment | Outcome and Take-home message | Risk of Bias Score |
|---|---|---|---|---|---|---|---|
| 26 | Olney et al, 2019, Burundi | 4-arm cluster-RCT repeated cross-sectional design including food rations, health services strengthening and promotion, and behaviour change communication on nutrition, hygiene, and health practices. | Enablement, Education, Persuasion. | Pregnant women (>4 months of gestation) and mothers of children aged 6-24 months from Burundi. | **Nutrition:** Three repeated cross-sectional surveys were conducted including household dietary diversity, hunger, food insecurity, and food consumption. Also mother diet diversity and child index of infant and young children feeding (6946 households). | At first follow-up, Tubaramure positively affected language but not motor development among children aged 4–23.9 months. Among infants 12–23.9 months, the programme positively affected language and motor development. At second follow-up, among children aged 24–41.9 mo, Tubaramure marginally affected motor development Significant positive impacts on diet, health, and nutritional indicators of children aged 12–23.9 months and health and nutritional indicators of children aged 24–29.9 months, supporting the plausibility of programme impacts on child development. | Low 6 |
| 27 | Leroy et al, 2018, Burundi | 4-arm cluster-RCT repeated cross-sectional design including food rations, health services strengthening and promotion, and behaviour change communication on nutrition, hygiene, and health practices. | Enablement, Education, Persuasion. | Pregnant women (>4 months of gestation) and mothers of children aged 6-24 months from Burundi. | **Nutrition:** Three repeated cross-sectional surveys were conducted including household dietary diversity, hunger, food insecurity, and food consumption. Also mother diet diversity and child index of infant and young children feeding (6946 households). height-for-age z score HHS, HFIAS, MDD. | Stunting (height-for-age z score) increased markedly from baseline to follow-up, but Tubaramure had a significant beneficial effect. Secondary analyses showed that the effect was limited to children whose mother and head of household had some primary education and who lived in households with above-median assets. | Low 6 |
| 28 | Leroy et al, 2016, Burundi | 4-arm cluster-RCT repeated cross-sectional design including food rations, health services strengthening and promotion, and behaviour change communication on nutrition, hygiene, and health practices | Enablement, Education, Persuasion. | Pregnant women (>4 months of gestation) and mothers of children aged 6-24 months from Burundi. | **Nutrition:** Three repeated cross-sectional surveys were conducted including household dietary diversity, hunger, food insecurity, and food consumption. Also mother diet diversity and child index of infant and young children feeding (6946 households). | Haemoglobin decreased and anaemia increased markedly from baseline to follow-up. Significant beneficial effect on both children and mothers who had given birth in the previous 3 months. Significant (P < 0.05) impacts on factors along the hypothesized impact pathways: dietary diversity, consumption of iron-rich foods, morbidity, and fever for child haemoglobin and dietary diversity, consumption of iron-rich foods, and current bed-net use for maternal anaemia. | Low 6 |
| 29 | Mulualem et al, 2016, Ethiopia | Cluster-RCT of Health Belief Model based education on complimentary feeding practices of mothers. | Education, Training, Incentivisation. | 160 households with children 6–18 months of age were selected by a systematic sampling method from 21 rural Kebeles of Ethiopia. | **Nutrition** Interviewing using questionnaires for mothers about feeding practice of the infants and children. Complementary feeding practices were measured by the following indicators: meal frequency; the proportion of infants and young children consuming a variety of food group for the previous 24 hours; and MDD. <br><br>**Psychosocial** Health belief model constructs were tested using a questionnaire (10 items). | Significant improvements in children's mean weight, weight for height, and weight for age occurred in the intervention site only. The KAP of mothers on pulse-incorporated complementary feeding and the nutritional status of their young children measured in terms of wasting and underweight prevalence were shown to be improved through an action-oriented and recipe-based nutrition education intervention. | Low 6 |
| 30 | Heckert et al, 2020, Burundi | Economic evaluation on a 4-arm cluster-RCT repeated cross-sectional design including food rations, health services | Enablement, Education, Persuasion. | Pregnant women (>4 months of gestation) and mothers of children aged 6-24 months from Burundi. | Total and per beneficiary cost, conducted cost consequence analyses, and estimated the cost savings from extending the programme for 2 years. | Providing food assistance for the full 1,000 days led to the lowest cost per percentage point reduction in stunting. Reducing the duration of ration eligibility reduced per beneficiary costs but was less effective. | Medium 4 |

*(Continued)*

**Table 3.** (Continued)

| | Author, publication year, country | Study design and intervention | Behaviour change wheel intervention function | Setting and participants | Assessment | Outcome and Take-home message | Risk of Bias Score |
|---|---|---|---|---|---|---|---|
| | | strengthening and promotion, and behaviour change communication on nutrition, hygiene, and health practices. | | | | A 2-year extension could have saved 18% per person. Programmes providing smaller rations or rations for shorter durations, although less expensive per beneficiary, may not provide the necessary dose to improve (biological) outcomes. Delivering effective programme for longer periods can generate cost savings by dispersing start-up costs and lengthening peak operating capacity. | |
| 31 | Ogunsile & Ogundele, 2016, Nigeria | Pre-test–Post-test quasi-experimental non-equivalent groups design. Nutrition education, using a board game. This was evaluated using questionnaires about knowledge of healthy eating. | Environmental restructuring, Enablement, Training. | One hundred and forty-three secondary school. Students selected from 4 local government areas in Ibadan, Nigeria. 143 (84 males and 59 females) adolescents with a mean age of 13.59 years. | **Nutrition** Questionnaires: Adolescents' Knowledge of Healthy eating Questionnaire, Adolescents' Attitude to Healthy Eating Questionnaire, Adolescent's Practice of Healthy Eating Questionnaire. (n=143) | Significant improvements on adolescents' knowledge, attitude and practice of healthy eating. Creative strategies should always be incorporated into nutrition education programme for adolescents. | Medium 4 |
| 32 | Tariku et al 2015, Ethiopia | Cluster-RCT of Health Belief Model based education on complimentary feeding practices of mothers. | Education, Training, Incentivisation. | 166 households with children 6–18 months of age were selected by a systematic sampling method from 21 rural Kebeles of Ethiopia. | **Nutrition** Interviewing using questionnaires for mothers about feeding practice of the infants and children. Complementary feeding practices were measured by the following indicators: meal frequency; the proportion of infants and young children consuming a variety of food group for the previous 24 hours; and MDD. **Psychosocial** Health belief model constructs were tested using a questionnaire (10 items). | Diet diversity significantly increased, including improvements in food groups were most noticeable as legumes & nuts. Intervention based on theory are successful at improving complimentary feeding. | Medium 4 |
| 33 | Galasso et al, 2009, Madagascar | Large scale community-based RCT nutrition programme that included weighing infants, providing nutrition counselling to the mothers, promoting behavioural change nutrition and hygiene education sessions, and suppling micronutrient supplementation. | Training, Education, Enablement. | Mothers and infants in 1999 in four provinces and expanded to all six provinces in Madagascar in 2000. | **Nutrition** Child LAZ and WLZ. Five tablet-based questionnaires administered at the three timepoints on household demographics, mother and primary caregiver questionnaire, child questionnaire, community health worker questionnaire and village details. | The programme helped 0–5 years old children in the participating communities to bridge their gap in weight-for-age z-score and the incidence of underweight. The programme had significant effects in protecting long-term nutritional outcomes (height-for-age z-scores and incidence of stunting). Less educated mothers and worse-off households were less placed to benefit from the programme in terms of nutritional outcomes. | Medium 4 |
| 34 | Saaka et al, 2021, Ghana | Two-arm, quasi-experimental, with pre- and post-test observations to evaluate radio health/nutrition education intervention in Ghana. | Persuasion, Modelling, Education. | 712 mothers with children aged 6–36 months from Northern Ghana. | **Nutrition** Anthropometric measures were converted to indices of length-for-age (LAZ), weight-for-age (WAZ) and weight-for length (WLZ) minimum meal frequency (MMF), the minimum dietary diversity (MDD) and the minimum acceptable diet (MAD). **Psychosocial** Nutrition knowledge, attitudes and practices (KAP) | Minimum dietary diversity and the minimum acceptable diet improved significantly. Nutrition related knowledge, attitudes and practices scores were significantly higher The intervention did not have significant effects on the nutritional status as measured by height-for-age Z-score or weight-for-height Z-score | Medium 3 |

*(Continued)*

**Table 3.** (Continued)

| | Author, publication year, country | Study design and intervention | Behaviour change wheel intervention function | Setting and participants | Assessment | Outcome and Take-home message | Risk of Bias Score |
|---|---|---|---|---|---|---|---|
| 35 | Yetnayet et al 2017, Ethiopia | A pre-test post-test controlled intervention study nutrition education intervention based on the Health Belief Model, and using pulses, to improve knowledge, attitude and practice of women of reproductive age. | Enablement, Training, Education | 200 randomly selected women from Southern Ethiopia. All women of reproductive age (15 – 49 years). | Food frequency questionnaire with constructs from the Health Belief Model and Knowledge, Attitude and practice items (30 items). | Significant improvement in the mean knowledge, attitude, and practice scores in the intervention group compared to control group. Significant improvement in the scores of Health Belief Model constructs. The success of this intervention may be due to using pulses processing and recipes demonstrations repeatedly, involving peer learning and experience sharing. | Medium 3 |
| 36 | Hurley et al, 2021, Malawi | Impact evaluation quasi-experimental, longitudinal nutrition programme which gave lipid-based nutrient supplement to infants and social and behaviour change communication to caregivers. | Enablement, Persuasion, Education. | Infants from 6–23 months of age, accompanied by caregivers in rural, agrarian districts in central Malawi. | **Nutrition** Infant LAZ, WLZ, and MUAC measurements. Child morbidity and handwashing practices. IYCF variables included continued breastfeeding, dietary diversity, minimum dietary diversity (MDD), minimum meal frequency (MMF), and minimum acceptable diet (MAD). | Growth velocities favoured programme children, such that LAZ (+0.12/y), WLZ (+0.12/y), and MUAC (+0.24 cm/y) measurements increased. Significant 13.8 pp reductions in the prevalence of malaria and fever at the 18-month follow-up. Significant improvements of 20 pp in minimum dietary diversity and minimum acceptable diet were seen in the programme versus comparison district at 18 months of follow-up. | Medium 2 |
| 37 | Lagerkvist et al, 2018, Nigeria | Cross-sectional design. Four-week intervention and field experiment providing a meal based on orange-fleshed sweet potato, rich in pro-vitamin linked to goal setting. | Education, Training, Enablement. | 556 randomly selected children aged 7–12 in Osun state, Nigeria. | **Nutrition** A survey questionnaire was administered to obtain baseline conditions before the nutrition education intervention session on food poverty and breakfast routines. Recall data on food intake focusing number of times main food groups were consumed per day during a typical week. **Psychosocial** After eating each participant rated how their school meal made them feel. Feelings were captured using two measures on a ve-point Likert scale with anchors (1=tired; 5=refreshed, and 1=sleepy; 5=awake) (n=556). | Planning by stating intentions increased the amount eaten. Priming of the experimental goals evoked positive feelings after eating. School meals programme should be designed to better align personal motivation with behavioural change in relation to dietary health. | Medium 2 |
| 38 | De Villiers et al, 2016, South Africa | A three-year cluster RCT at primary schools in low socioeconomic settings. Action planning process with teachers to identify their own priorities, provided with South African food-based dietary guidelines. Children completed a questionnaire on nutrition knowledge, self-efficacy and behavioural items. | Training, Modelling, Education. | Participants were year four children, 9 years old (n=500) at eight schools in the intervention group and eight in the control group (n=498), in the Western Cape Province of South Africa. | **Nutrition and Psychosocial** Knowledge, Self-Efficacy and Behaviour (KAP) Questionnaire (n=717). Nutrition knowledge had 29 items, nutrition behaviour has 8 items and self-efficacy had 12 items. | The intervention improved nutrition knowledge and self-efficacy significantly in primary school children, but it did not improve their eating behaviour. A programme mostly driven by school staff and with specific guidelines on how to integrate it with the curriculum could improve nutrition knowledge and self-efficacy in children. | Medium 2 |
| 39 | Steyn et al, 2015, South Africa | A three-year cluster RCT at primary schools in low socioeconomic settings. Action planning process with teachers to identify their own priorities, provided with South African food-based dietary guidelines. Children completed a questionnaire on nutrition knowledge, self-efficacy and behavioural items. | Training, Modelling, Education. | Participants were year four children, 9 years old (n=500) at eight schools in the intervention group and eight in the control group (n=498), in the Western Cape Province of South Africa. | **Nutrition and Psychosocial** Knowledge, Self-Efficacy and Behaviour (KAP) Questionnaire (n=717). Nutrition knowledge had 29 items, nutrition behaviour has 8 items and self-efficacy had 12 items. | The estimated dietary diversity score intervention effect over the two years was not significant. Food groups least consumed were eggs, fruit and vegetables. Unhealthy snack consumption in terms of frequency of snack items consumed did not improve significantly in intervention or control schools. | Medium 2 |

*(Continued)*

**Table 3.** (Continued)

| | Author, publication year, country | Study design and intervention | Behaviour change wheel intervention function | Setting and participants | Assessment | Outcome and Take-home message | Risk of Bias Score |
|---|---|---|---|---|---|---|---|
| 40 | Grant et al, 2022, Tanzania | Community-based cross-sectional survey to assess the association between monthly participation in community-level nutrition group meetings implemented within the VISTA project (a community-based nutrition education and SBCC intervention to promote the incorporation of biofortified orange fleshed sweet potato into children's diets) on caregiver health and nutrition knowledge and practices. | Education, Enablement, Persuasion. | Primary caretakers of children aged 6-59 months in households in eastern and southern highland zones of Tanzania (all seven VISTA Tanzania project intervention districts). | Indices for caregivers' knowledge of nutrition, health and childcare, household (HDD) and young child dietary diversity (CDD), and vitamin A (VA) intakes. | Participation in nutrition group meetings was significantly associated with the health and childcare knowledge score (HKS), HDD and CDD scores, and household and young child VA intake; the magnitude of the associations was greater for caregivers who attended at least four meetings. Findings emphasize the need for programs that seek to address the issues present in the use of nutrition SBCC at the community level to improve maternal or caregiver knowledge and practices and subsequently the nutrition status of infants and young children | Medium 1 |
| 41 | Tomlinson et al, 2020, South Africa | RCT to promote maternal sensitivity and maternal–child attachment related to improve infant feeding. | Training, Enablement, Modelling. | 449 women assessed during pregnancy, and then when their infant was 2, 6, 12 and 18months old in a peri-urban area with high levels of poverty, near Cape Town, South Africa. | **Nutrition** Mothers were filmed and observed feeding their baby for 5 minutes. Recordings were rated on five dimensions of maternal sensitivity and on maternal Intrusiveness. **Other** Mothers were interviewed to collect demographic details, living circumstances and obstetric, delivery and post-birth information, including whether or not they were breastfeeding at 2 and 6months postnatally Mental health assessment to determine level of depressive symptoms. | During a feeding interaction, maternal sensitivity was significantly improved among non-breastfeeding mothers who received the intervention. Particularly, maternal responsiveness to infant cues and synchronous interactions was higher among non-breastfeeding intervention mothers compared to control group mothers. The results also show that non-breastfeeding mothers who received the intervention were significantly less intrusive in their interactions with their infants. This suggest that the intervention offered a protective effect for non-breastfeeding mothers. | Medium 1 |
| 42 | Mutiso et al, 2018, Kenya | Cross-sectional study to examine the effect of nutrition education and psychosocial on the use of comprehensive IYFC practices among different women. | Education, Persuasion, Enablement. | 665 women of different categories: (1) pregnant women who were also caregivers of children below 2 years, (2) mothers of 0–5 months old children, (3) mothers of 6–23 months old children, and (4) potential mothers in Kenya. | **Psychosocial** Four constructs based on the Theory of Planned Behaviour to proxy psychosocial factors: attitudes, subjective norms, PBC, and knowledge. | Nutrition education participation mother-to-mother club health talks had the greatest effect on the extent to which IYCF practices are used, likely because these platforms provide opportunity to discuss, and overcome, some of barriers with peers within the communities. This study also finds strong evidence that these nutrition education strategies jointly affect the expected number of IYCF practices used. | Medium 1 |
| 43 | Workicho et al, 2021, Ethiopia | Quasi-experimental design social and behaviour change communication in Ethiopia. | Modelling, Persuasion, Education. | Pregnant and/or lactating women from Amhara and Oromia regions of Ethiopia. | **Nutrition** Minimum Dietary Diversity (MDD) Minimum Meal Frequency (MMF) Minimum Acceptable Diet (MAD) Women's diet diversity was based on a 24-h qualitative recall. | 13.6% change in iron folic acid (IFA) intake for 3 months. Not statistically significant but large to moderate positive changes in child minimum diet diversity (20%), minimum acceptable diet (18%) and women diet diversity (7.9%). | Medium -1 |

(*Continued*)

**Table 3.** (Continued)

| | Author, publication year, country | Study design and intervention | Behaviour change wheel intervention function | Setting and participants | Assessment | Outcome and Take-home message | Risk of Bias Score |
|---|---|---|---|---|---|---|---|
| 44 | Skar et al, 2019, Mozambique | Psychosocial support intervention based on the International Child Development Programme integrated with a health and nutritional supplement. | Training, Enablement, Modelling,. | 350 children aged 1–18years were recruited from 12 child centers for orphans and other vulnerable children in the Maputo area, Mozambique. | **Nutrition** Age, gender, height, and weight of the children at the intervention centers were measured before and after the intervention. **Psychosocial** Child-Reported Strengths and Difficulties questionnaire. Professional caregivers from the intervention centers completed an open-ended response survey about their work at the centers and whether they noticed any changes in themselves, the children at the centers, or at the child centers in general after receiving the integrated programme. | Improvement among 93.5 percent of the children categorized as malnourished before the intervention. Children's self-reported data on strengths and difficulties (N = 79) suggest significant increases in prosocial behaviour in the intervention group. Caregivers (N = 40) reported strengthened relationships, improved communication, and improved nutritional menus within the intervention centers. | Medium -2 |
| 45 | Ruel-Bergeron et al, 2019b, Malawi | Process evaluation quasi-experimental study design, with cross-sectional baseline. Daily, fortified, small-quantity lipid-based nutritional Supplement and behaviour change messages around optimal infant and young child feeding (IYCF) and water, sanitation, and hygiene. | Training, Persuasion, Enablement. | Mother and their children aged 6–23 months in two neighbouring, rural districts in Malawi. | Programme monitoring and evaluation data were used to measure programme recruitment, reach, and fidelity. Structured direct observations and knowledge questionnaires with programme volunteers measured quality aspects of programme fidelity. The number of times activities were done correctly was used to tabulate proportions used to represent programme functioning. | Half (49.5%) of eligible children redeemed programme benefits by 8 months of age during th first 4 years of programme implementation. Knowledge of IYCF, WASH, and SQ-LNS messages by volunteers was >85% for most messages, except ability to list the 6 food groups (35.7%). Structured direct observations of SQ-LNS distributions indicated high fidelity to programme design, whereas those of household-level counselling sessions revealed lack of age-appropriate messaging. Programme reach showed participation in monthly distribution sessions of 81.0%, group counselling of 93.3%, and individual-level counselling of 36.9%. | Medium -3 |
| 46 | Ezezika et al, 2018, Nigeria | Qualitative evaluation of a pilot study on gamification of nutrition through board games, clubs and vouchers. | Enablement, Training, Education. | Three secondary schools in Abuja, Nigeria. 31 adolescents 13-17 years participated in the focus groups. | Semi-structured focus groups were conducted with 31 participants about their perceptions of the intervention and how it influenced their eating behaviour, attitudes and knowledge about nutrition. | Participants perceived that the intervention shifted their perceptions and preferences, and altered their behaviour by eating nutritious foods, increasing physical activity, and improving overall well-being. Gamification has short-term positive influences on adolescent dietary and physical activity behaviours, but more research needs to explore the long-term improvements. | High -4 |
| 47 | Fernandes et al, 2016, Ghana | Evaluation of the School Meals Planner Package and behaviour change communication campaigns introduced in Ghana. Developing menus for home-grown school feeding programmes. | Education. Enablement, Persuasion. | Intervention introduced in 42 districts in Ghana, reaching more than 320 000 children. | Monitoring visits were undertaken in all the 42 pilot districts at least once every school term, where they asked teachers and caterers about different aspects of the intervention. | Monitoring and evaluation approaches found positive findings including that the tool was easy to use but called for more capacity building. Complaints included that correct ingredient usage estimates made meals too expensive. The tool is valued by Ghana's government at the highest levels as it is adopted as an official policy, due to supporting nutritious, locally sourced menus for the school feeding programme in Ghana. | High -4 |
| 48 | Ruel-Bergeron et al, 2019a, Malawi | Monitoring and evaluation quasi-experimental study design, with cross-sectional baseline. Daily, fortified, small-quantity lipid-based nutritional | Training, Persuasion, Enablement. | Mother and their children aged 6–23 months in two neighbouring, rural districts in Malawi. | **Nutrition** Anthropometric measurements: children's length, MUAC, and head circumference, BMI, LAZ, WLZ. | The absence of inclusion of an older cohort of children (24–41 months) in the impact evaluation limited ability to measure programme impact on children who had received full exposure. Ongoing training, motivation, supervision, and | High -6 |

*(Continued)*

**Table 3.** (Continued)

| | Author, publication year, country | Study design and intervention | Behaviour change wheel intervention function | Setting and participants | Assessment | Outcome and Take-home message | Risk of Bias Score |
|---|---|---|---|---|---|---|---|
| | | Supplement and behaviour change messages around optimal infant and young child feeding (IYCF) and water, sanitation, and hygiene. | | MMF, MDD. Caregivers' IYCF knowledge related to breastfeeding, complementary feeding, and feeding during illness was assessed and a knowledge score calculated. | | encouragement of care group volunteers is critical to programme success and a key element for high retention of participants throughout the life of the programme. Investing in understanding the country and programme context, including the behavioural patterns of the population is necessary for implementing the most efficient M&E systems and reaching the target population. | |
| 49 | Christian et al, 2020, Malawi | Impact evaluation quasi-experimental study design, with cross-sectional baseline. Daily, fortified, small-quantity lipid-based nutritional supplement and behaviour change messages around optimal infant and young child feeding (IYCF) and water, sanitation, and hygiene. | Training, Persuasion, Enablement. | Mother and their children aged 6–23 months in two neighbouring, rural districts in Malawi. | **Nutrition** Anthropometric measurements: children's length, MUAC, and head circumference, BMI, LAZ, WLZ. MMF, MDD. Caregivers' IYCF knowledge related to breastfeeding, complementary feeding, and feeding during illness was assessed and a knowledge score calculated. | No differences in mean LAZ or prevalence of stunting were found at end line. However, mean weight, WLZ, and MUAC were higher at end line by 150 g, 0.22, and 0.19 cm, respectively, in the programme compared with the comparison district (all P < 0.05). Weekly reports of high fever and malaria were also lower by 6.4 and 4.7 percentage points, respectively, in the programme compared with the comparison district. There was no impact on anaemia. Children's dietary diversity score improved by 0.17, and caregivers' infant and young child feeding and hand-washing practices improved by 8–11%in the programme compared with the comparison district. | High -7 |
| 50 | Muehlhoff et al, 2017, Malawi (and Cambodia) | Process review (focus groups and interviews) on an intervention that provided agricultural support and community-based nutrition education on improved infant and young child feeding. | Incentivisation, Education, Environmental restructuring. | Focus groups in Malawi were conducted in selected villages involving a total of 170 stakeholders – mothers, caregivers, project stakeholders and government staff. | Interviews and focus groups conducted guided by 60 questions on agriculture-nutrition linkage, community-based nutrition education activities, family environment, socio-cultural factors influencing infant and young children feeding and involvement of service providers. | Barriers included availability and access to nutritious, affordable foods, and adverse effects on mothers' time and childcare needs. Integrated programme can ensure that caregivers with young children benefit from multiple sectors' work. Volunteers are a crucial resource for community-based interventions. | High -7 |
| **4 BCW IF** | | | | | | | |
| 51 | Kim et al, 2019a, Ethiopia | A pre- and post-intervention adequacy evaluation design was used; repeated cross-sectional surveys, of cluster-RCT evaluation design with cross-sectional surveys. Alive & Thrive provided intensive behaviour change interventions through 4 platforms: interpersonal communication (IPC), nutrition-sensitive agricultural activities (AG), community mobilization (CM), and mass media (MM). | Persuasion, Environmental restructuring, Modelling, Enablement. | Households with children aged 6–23.9 months in Ethiopia. | **Nutrition** Maternal recall of all foods and liquids given to children in the first few days after birth and in the 24 h prior to the survey. Household food security was measured using FANTA/USAID's Household Food Insecurity Access Scale. | Significant differential declines in stunting prevalence were observed in children aged 6–23.9 months, decreasing from 36.3% to 22.8% in the intensive group. Dose–response analyses showed higher odds of MDD and MMF and higher HAZ among women exposed to 3 or 4 intervention platforms. Strong relation between agricultural intervention and egg consumption, which led to increased child dietary diversity and HAZ. | Low 8 |
| 52 | Byrd et al, 2019, Kenya | Quantitative evaluation of nutrition, water and sanitation cluster-RCT. Community based promoters counselled | Enablement, Training, Education, Persuasion. | 10 districts within in Kenya, with one to two villages with a minimum of six women per cluster. | Household interviews conducted at baseline, year 1 and year 2. | Compared to controls, there were no differences for MDD, MMF, and MAD for nutrition counselling or supplementation arms. To improve complimentary feeding practices, a higher intensity and frequency of behaviour change communication may see more sustained results. | Low 7 |

(*Continued*)

**Table 3.** (Continued)

| | Author, publication year, country | Study design and intervention | Behaviour change wheel intervention function | Setting and participants | Assessment | Outcome and Take-home message | Risk of Bias Score |
|---|---|---|---|---|---|---|---|
| | | households on optimal infant and child feeding practices, and children 6–24 months of age were provided with small quantity lipid based nutrient supplements. | | **Nutrition** Infant and young child feeding survey was administered after year 1 (n=3652) and 2 (n=4987). An in-home quantitative 24 hour recall survey of foods (MDD, MMF, MAD) was conducted (n=185), and energy and nutrient intakes were calculated by linking intakes with WorldFood Dietary Assessment. | | | |
| 53 | Mlinda et al, 2018, Tanzania | RCT facility-based intervention practical nutrition programme on feeding skills in caregivers of children with cerebral palsy. | Education, Training, Enablement, Environmental restructuring. | 110 children under 5 with cerebral palsy in Dar es Salaam, Tanzania. | Researchers observed caregiver looking after the infant and caregiver reported on their stress levels during feeding (n=228). | The intervention significantly improved feeding skills of caregiver such as feeding children slower and involved the child during the feeding process. Strengthening nutrition education and services for caregivers of children with cerebral palsy improves the health outcomes of children and reduce stress among parents/caregivers. | Low 7 |
| 54 | Kumar et al, 2018, Zambia | Cluster RCT "Realigning Agriculture for Improved Nutrition" RAIN intervention in a multisectoral approach integrating agricultural diversification, promotion of gender equality, women's empowerment and nutrition behaviour change communication. | Modelling, Education, Enablement, Persuasion. | Mother's groups who were pregnant or lactating in rural Zambia. | **Nutrition** Child practices were measured using maternal recall of practices related to breastfeeding and complementary feeding using eight indicators recommended by the WHO. Infancy and young children feeding knowledge was assessed based on the mothers' answers to a series of questions related to knowledge on established IYCF practices. WHZ scores. **Psychosocial** Women's empowerment was measured aligned with the domains included in the Women's Empowerment in Agriculture Index. | Positive effects on women's empowerment, infant and young child feeding knowledge, child morbidity and WHZ scores, but no impact on stunting. Fostering higher participation rates could support greater impacts on child nutrition outcomes. | Low 5 |
| 55 | Rosenberg et al, 2018, Zambia | Cluster RCT "Realigning Agriculture for Improved Nutrition" RAIN intervention in a multisectoral approach integrating agricultural diversification, promotion of gender equality, women's empowerment and nutrition behaviour change communication. | Modelling, Education, Enablement, Persuasion. | Mother's groups who were pregnant or lactating in rural Zambia. | **Nutrition** Child practices were measured using maternal recall of practices related to breastfeeding and complementary feeding using eight indicators recommended by the WHO. Infancy and young children feeding knowledge was assessed based on the mothers' answers to a series of questions related to knowledge on established IYCF practices. WHZ scores. **Psychosocial** Women's empowerment was measured aligned with the domains included in the Women's Empowerment in Agriculture Index. | The programme increased diversity in crops grown and the number of months in which various food groups were harvested. The programme substantially increased the percentage of households producing three nutritious crops it promoted (groundnuts, rape and tomatoes). Modest increases in household access to diverse food groups. Despite modest increases in the proportion of children consuming pulses, legumes and nuts, ultimately there were no significant improvements in the overall dietary diversity of young children or their mothers. | Low 5 |

*(Continued)*

**Table 3.** (Continued)

| | Author, publication year, country | Study design and intervention | Behaviour change wheel intervention function | Setting and participants | Assessment | Outcome and Take-home message | Risk of Bias Score |
|---|---|---|---|---|---|---|---|
| 56 | Flax et al, 2021, Rwanda | Cluster-randomized controlled trial designed to test the impact of a Social and Behaviour Change Communication (SBCC) intervention to promote the consumption of Animal source foods (ASFs). | Enablement, Education, Training, Persuasion. | The mother was 18–49 years of age and had a child who was 12–29 months of age Nyabihu and Ruhango Districts, Rwanda. | **Nutrition** World Health Organization (WHO) infant and young child feeding questionnaire (24-hour recall) Maternal Animal source foods knowledge questions. | More mothers in the intervention group compared with the control group knew they should feed their children Animal source foods and give them 1 cup of cow's milk per day. Children's consumption of fresh cow's milk 2 or more times per week increased in the intervention group, although not significantly and minimum dietary diversity was unchanged. In poor households receiving a livestock transfer, strategies to further tailor SBCC and increase cow's milk production may be needed to achieve larger increases in children's frequency of milk consumption | Medium 4 |
| 57 | Katenga-Kaunda et al, 2021, Malawi | Cluster randomised controlled trial supplementary nutrition education, dietary counselling and routine ANC services in Malawi. | Modelling, Enablement, Education, Persuasion. | 257 pregnant women from Namkumba area in Mangochi, one of the 12 districts in the southern region of Malawi. | **Nutrition** 24 h quantitative food consumption Dietary diversity score Minimum dietary diversity for woman.<br><br>**Psychosocial** Nutrition knowledge questionnaire | Significant improvements in nutrition knowledge, dietary diversity and nutrition behaviour in the intervention group compared with controls. Increased nutrition knowledge was associated with improved dietary diversity only in the intervention, who also improved their nutrition perceptions and behaviour. Antenatal nutrition education needs strengthening to improve dietary intakes in pregnancy in this low resource-setting. | Medium 4 |
| 58 | Briaux et al, 2020, Togo | Parallel-cluster–RCT on unconditional cash transfer (UCTs) with behaviour change communication. | Persuasion, Incentivisation, Enablement, Training. | Women > 3 months pregnant and mothers of children aged 0–23 months 5 districts in Burkina Faso. | **Nutrition** Anthropometric measurements HAZ scores. Qualitative multiple-pass 24-h recall using the MDD-W.<br><br>**Other** WHO's Violence Against Women instrument (VAWI), Household Budget Survey (HBS) | UCTs had a protective effect on HAZ scores, which deteriorated in the control arm while remaining stable in the intervention arm but had no impact on stunting. UCTs positively impacted both mothers' and children's (18–23 months) consumption of animal source foods and household food insecurity. The UCT arm did not impact on reported child morbidity 2 week's prior to report but reduced the financial barrier to seeking healthcare for sick children. Women who received cash had higher odds of delivering in a health facility and lower odds of giving birth to low-birth-weight babies. Positive effects were also found on women's knowledge and physical intimate partner violence. | Medium 4 |
| 59 | Lion et al, 2018, Nigeria | Quasi-experimental pre- and post-intervention design with a randomly assigned intervention and control on a behaviour change programme and the use of green leafy vegetables (greens) and iron-fortified bouillon cubes in stews to improved iron intake. | Enablement, Education, Persuasion, Incentivisation. | 527 mother-daughter pairs (daughters aged 12–18) from Ile-Ife (Intervention town). Osogbo (Control town) in Nigeria. | **Nutrition** FFQ was created and tailored to the Nigerian context. The questionnaire captured how often participants had prepared different types of stews in the past 2 weeks, and which ingredients were used, including quantity. Quantity assessments were based on locally used units. An intake question was used.<br><br>**Psychosocial** The determinants of behaviour questionnaire measured the two key behaviour of interest: "adding Knorr iron-fortified cubes to beef stew" and "adding green leafy vegetables to beef stew". The questionnaire | Change in iron-fortified cubes added to stews did not significantly differ between towns. Increase in cubes added to soups was significantly larger in the Intervention town compared to the Control Town. Change in greens added to soups was significantly larger in the Control town compared to the Intervention town. The intervention positively influenced awareness of anaemia and the determinants of behaviour in the Intervention town, with hardly any change in the Control town. | Medium 4 |

*(Continued)*

**Table 3.** (Continued)

| | Author, publication year, country | Study design and intervention | Behaviour change wheel intervention function | Setting and participants | Assessment | Outcome and Take-home message | Risk of Bias Score |
|---|---|---|---|---|---|---|---|
| | | | | | included items measuring Attitude, Injunctive Social Norm, Descriptive Social norms, Perceived Behavioural Control, Habit and Intention. | | |
| 60 | Katenga-Kaunda et al, 2022, Malawi | Qualitative data from a cluster-randomised maternal education trial was used to conduct a thematic analysis using a social ecological framework to describe the factors that influenced dietary adherence | Modelling, Enablement, Education, Persuasion. | Ten pregnant women from the intervention arm of the original RCT, 22 significant family members to the women (husbands and mothers-in-law) from the same intervention communities and twelve counsellors who had conducted the educational trainings and who had some influence in their local communities Nankumba area in Mangochi, Southern Malawi | Structured, individual in-depth interviews with the pregnant women and focus group discussions with significant family members and the counsellors two months before the intervention ended. In addition, observational data recorded by the local team members during the intervention were included. | Main barriers of adherence to the intervention were taste, affordability, and poverty Use of powders and one-pot dishes, inclusion of both women and significant family members and a harmonisation with local food practices enabled adherence to the intervention It is crucial to focus the dietary education and counselling intervention on locally available ingredients and food processing methods Use of contextualised food-based solutions to combat maternal malnutrition was observed to be relatively cheap and sustainable | Medium 3 |
| 61 | Katenga-Kaunda et al, 2020, Malawi | Evaluation of a cluster RCT where the intervention group received nutrition education and dietary counselling with regard to improvements in quality of diet and use of the Theory of Planned Behaviour to explain changes in their dietary behaviour | Modelling, Enablement, Education, Persuasion. | 257 pregnant women in twenty villages in the Namkumba area in Mangochi, Southern Malawi | DDS, Malawian six food groups (SFG) score Affective attitudes were assessed as women's self-evaluation at baseline and at study end point Perceived difficulty was measured through assessment of nutrition skills To assess behaviour controllability, the women were asked if they experienced challenges with sustaining a diverse diet Subjective norm was assessed as individual v. family involvement in sustaining a diverse diet | The intervention achieved improvements in DDS and the Six Food Group Pyramid (SFG) score, especially in intakes of micronutrient-rich foods The theorised behaviour mediators (i.e., nutrition attitudes, nutrition behaviour control and subjective norm) that had improved were significantly associated with high DDS Attainment of high DDS was a consequence of the women's belief in the effectiveness of the proposed nutrition recommendations | Medium 3 |
| 62 | Kim et al, 2016, Ethiopia | A pre- and post-intervention adequacy evaluation design was used; repeated cross-sectional surveys, of cluster-RCT evaluation design with cross-sectional surveys. Alive & Thrive (A&T) provided intensive behaviour change interventions through 4 platforms: interpersonal communication (IPC), nutrition-sensitive agricultural activities, community mobilization, and mass media. | Persuasion, Environmental restructuring, Modelling, Enablement. | Households with children aged 6–23.9 months in Ethiopia. | **Nutrition** WHO-recommended core IYCF indicators related to BF and CF (for children 0–24 months of age) and stunting in children 24–59.9 months of age. Weight and length/height were converted into height-for-age z-scores (HAZ), weight-for-age z-scores (WAZ), and weight-for-height z-scores (WHZ) according to the WHO child growth standards | Early BF initiation and exclusive BF increased by 13.7 and 9.4 percentage points. Complementary foods, minimum dietary diversity, minimum meal frequency increased significantly in the intensive group but remained low at end line. Timely introduction and intake of foods promoted by the interventions improved significantly, but anthropometric outcomes did not. | Medium 3 |

(*Continued*)

**Table 3.** (Continued)

| | Author, publication year, country | Study design and intervention | Behaviour change wheel intervention function | Setting and participants | Assessment | Outcome and Take-home message | Risk of Bias Score |
|---|---|---|---|---|---|---|---|
| 63 | Flax et al, 2022, Nigeria | Pre-post evaluation of a complementary feeding SBCC intervention which engaged parents through community meetings, religious services, home visits from community health extension workers, mobile phone messages (fathers only), and mass media | Enablement, Education, Training, Persuasion. | Fathers and mothers with a child aged 6–23 months in 6 of 12 wards in the Igabi local government area of Kaduna State, Nigeria | Baseline and endline cross-sectional population-based surveys<br><br>Complementary feeding indicators measured as part of the mother's questionnaire using the WHO IYCF questionnaire (MMD, specific food groups, MMF, and MAD)<br><br>Fathers' and mothers' complementary feeding knowledge, fathers' support for complementary feeding, and mothers' perceptions of fathers' support | Children's MDD did not change, children's consumption of fish and eggs and MMF increased<br>The intervention improved complementary feeding practices, fathers' and mothers' knowledge of complementary feeding, and fathers' support for complementary feeding, despite low levels of reported exposure | Medium 2 |
| 64 | Jacobs et al 2013, South Africa | Evaluate the effectiveness of the Making the Difference programme - an education and activity-based intervention for Grade 4 learners at primary schools. Programme included EduModules and educational visits to stores, suppliers and distribution centres, and talks for learners and parents by a network of trained dietitians. | Enablement, Education Training, Persuasion. | Four intervention and five control schools (n = 325 students) were selected in Western Cape, South Africa. | Learner-centred questionnaire, was developed as part of the Making the Difference programme school-based research programme, included constructs about knowledge, attitudes (self-efficacy) and practices in terms of nutrition and physical activity (n=325). | Small significant improvement in eating vegetables and taking lunch boxes to school but not explained by improvements in healthy eating, self-efficacy or knowledge.<br>As small significant results, the programme may need further evaluation to assess if this a worthwhile scheme. | Medium 1 |
| 65 | Kim et al, 2019b, Ethiopia | Endline survey data from impact evaluations of Alive & Thrive's intervention programs to improve IYCF practices were used to examine the extent of and factors associated with intervention exposure (IPC with or without other SBCC interventions [MM, CM, and nutrition-sensitive agricultural (AG) activities]) | Persuasion, Environmental restructuring, Modelling, Enablement. | Mothers with children aged <2 years in Bangladesh (n = 1001), Ethiopia (n = 1720), and Vietnam (n = 1001) | Face-to-face interviews using structured questionnaires<br><br>Four IYCF practice indicators: exclusive breastfeeding (EBF) to 6 months, MMF, MMD, and consumption of iron-rich or iron-fortified foods | In Ethiopia, exposure to IPC with other interventions was associated with higher odds of achieving MMF, MDD and consumption of iron-rich foods<br>Near-monthly visits were associated with 2–3 times higher odds of IYCF practices<br>Exposure matters for impact, but the combination of behaviour change interventions and number of IPC contacts required to support IYCF behaviour change are context specific | Medium -1 |
| 66 | Nordhagen and Klemm, 2018, Burkina Faso, Côte d'Ivoire, Senegal, Tanzania | Consecutive cross-sectional surveys conducted every 5 months among ~15% of households participating in Helen Keller International's model for nutrition-sensitive poultry production | Training, Environmental restructuring, Persuasion, Modelling. | Households in four diverse African contexts - three rural and one urban (Burkina Faso, Tanzania, Senegal, and Cote d'Ivoire) | Data (e.g., mortality of chickens, vaccination rates, and egg use) were collected by teams of trained external enumerators in the local language using a tablet-based questionnaire. The survey also included open-ended questions to ascertain participants' reasons for not applying certain project-promoted practices.<br><br>Simple recall questions were used to assess egg consumption in the last 7 days for young children and in the past 24h for women | Despite project-provided training and inputs, there was only limited uptake of many "best practices"<br>Egg consumption remained low; however, children whose mothers were exposed to project messages on nutrition were more likely to eat eggs, and consumption was consistently higher among households with chickens<br>Key lessons learned from implementation: (a) strong behaviour change communication is needed to encourage egg consumption, (b) nutrition-sensitive village poultry programmes should often focus more on improved practices than improved breeds, (c) supporting women's chicken production is not a route to empowerment without complementary activities that directly support women's ownership and decision making | Medium -3 |

(*Continued*)

**Table 3.** (Continued)

| | Author, publication year, country | Study design and intervention | Behaviour change wheel intervention function | Setting and participants | Assessment | Outcome and Take-home message | Risk of Bias Score |
|---|---|---|---|---|---|---|---|
| 67 | Kim et al, 2015, Ethiopia | Process evaluation of cluster-RCT evaluation design with cross-sectional surveys. Alive & Thrive provided intensive behaviour change interventions through 4 platforms: interpersonal communication (IPC), nutrition-sensitive agricultural activities, community mobilization, and mass media. | Persuasion, Environmental restructuring, Modelling, Enablement. | Households with children aged 6–23.9 months in Ethiopia. | Data from a qualitative study among three levels of front-line workers (n = 54), i.e. supervisors, health extension workers (HEWs), and community volunteers, and among mothers with children under two years of age (n = 60); and cross-sectional PE surveys with FLWs (n = 504) and mothers (n = 750) in two regions (Tigray and SNNPR) were analysed to examine programme fidelity. | Job aids were used regularly by most supervisors and HEWs, but only 54% of volunteers in Tigray and 39% in SNNPR received them. Quality of programme message delivery was lower among volunteers, and aided recall of key messages among mothers was also low Although FLW supervision exposure was high, content and frequency were irregular. | High -6 |
| 68 | Mukuria 2016, Kenya | Evaluation study on engaging fathers or grandmothers in improving mother child feeding practice. This included peer education dialogue groups with fathers and grandmothers. | Education, Incentivisation, Training, Modelling. | The selected intervention area for fathers and grandmothers in Western Kenya. 177 fathers and 156 grandmothers in intervention. For evaluation, four focus groups were held with fathers and four with grandmothers. 14 interviews were held with Ministry of Health officers, community and religious leaders, women's group leaders, and community health extension workers. | The process evaluation included in-depth interviews with father mentors (n=7) and grandmother mentors (n=10) and focus groups with fathers and grandmothers (8 and 10 FGDs, respectively). Baseline survey to assess knowledge and practices related to maternal nutrition and infant and young child feeding and social support provided to mothers by fathers and grandmothers about the care and feeding of young children. Post intervention survey assessed maternal, infant, and young child nutrition knowledge and practices; maternal and child dietary intake based on 24-hour and seven-day recalls; and (social support provided to mothers by fathers and grandmothers. | As the number of social support actions increased, the likelihood of a mother reporting that she had fed her infant the minimum number of meals in the past 24 hours also increased between baseline and end line. When taking into account the interaction effects of intervention area and increasing social support over time, a significant association was found in the grandmother intervention area on dietary diversity No significant effects were found on minimum acceptable diet. | High -9 |
| 69 | Thuita et al 2015, Kenya | Evaluation study on intervention on engaging fathers or grandmothers in improving mother child feeding practice. This included peer education dialogue groups with fathers and grandmothers. | Education, Incentivisation, Training, Modelling. | The selected intervention area for fathers and grandmothers in Western Kenya. 177 fathers and 156 grandmothers in intervention. For evaluation, four focus groups were held with fathers and four with grandmothers. 14 interviews were held with Ministry Of Health officers, community and religious leaders, women's group leaders, and community health extension workers. | The process evaluation included in-depth interviews with father mentors (n=7) and grandmother mentors (n=10) and focus groups with fathers and grandmothers (8 and 10 FGDs, respectively). Baseline survey to assess knowledge and practices related to maternal nutrition and infant and young child feeding and social support provided to mothers by fathers and grandmothers about the care and feeding of young children. Post intervention survey assessed maternal, infant, and young child nutrition knowledge and practices; maternal and child dietary intake based on 24-hour and seven-day recalls; and (social support provided to mothers by fathers and grandmothers. | Men typically provide advice on starchy foods, such as maize and maize meal, and animal protein while mothers and grandmothers often give advice on fruits and vegetables to families. Recommendations for future interventions should make use of existing community level structures allowing groups to select their leaders, and design interventions informed and guided by gender roles. Engaging father, grandmothers and mother would enhance the impact. | High -9 |
| | | | | | **5 BCW IF** | | |

*(Continued)*

**Table 3.** (Continued)

| | Author, publication year, country | Study design and intervention | Behaviour change wheel intervention function | Setting and participants | Assessment | Outcome and Take-home message | Risk of Bias Score |
|---|---|---|---|---|---|---|---|
| 70 | Gelli et al, 2020, Malawi | Evaluation of the one- year longitudinal cluster-RCT on early childhood development centre–based agriculture and nutrition intervention, including behaviour change communication training. | Enablement Education, Modelling, Persuasion, Environmental restructuring. | 60 community-based childcare centres, covering 1248 preschool children (aged 36–72 months) and 304 younger siblings (aged 6–24 months), in the Zomba district of Malawi. | **Nutrition** DDS, MDD, Knowledge of infant and young child feeding was assessed through caregiver interviews. WAZ, HAZ, and WHZ. | Positive impacts were found for several individual micronutrient intakes: vitamin A, vitamin C, riboflavin, zinc. These impacts were driven by effects on younger children (aged 3–4 years). Using a preschool platform to implement a nutrition-sensitive BCC intervention is an effective strategy to improve the adequacy of micronutrient intake of preschool children in food-insecure settings. | Low 10 |
| 71 | Gelli et al, 2018, Malawi | One year evaluation of a longitudinal cluster-RCT on early childhood development centre–based agriculture and nutrition intervention, including behaviour change communication training. | Enablement Education, Modelling, Persuasion, Environmental restructuring. | 1248 preschool children (aged 36–72 mo) and 304 younger siblings (aged 6–24 mo) from n 60 community-based childcare centers in the Zomba district of Malawi since 2008. | **Nutrition** Household food production to measure dietary diversity. Dietary assessment was undertaken using the interactive multi-pass 24-h recall method. Child dietary diversity (for preschoolers) and household dietary diversity were measured. Anthropometric data included measurements of height and weight for all children from 6 to 72 months.<br><br>**Psychosocial** Caregiver IYCF knowledge and practices interviews and questionnaires. | Compared with the control group, preschool children in the intervention group had greater increases in nutrient intakes and in dietary diversity. No impacts on anthropometric measures were seen in preschoolers. Younger siblings in the intervention group had greater increases in height-for-age z scores than did children in the control group and greater reductions in the prevalence of stunting. | Low 10 |
| 72 | Olney et al, 2016, Burkina Faso | Evaluation of 2-year RCT enhanced-homestead food production. This focuses on agricultural production and behaviour change communication by dedicating land to women's production and consumption of nutrient-rich foods and to generate additional income. | Education, Training, Persuasion, Environmental restructuring, Modelling. | 55 in eastern Burkina Faso with 55 all women with children 3–13months. | **Nutrition** Household consumption of individual food groups was examined by using detailed consumption data (7-d recall) on 57 commonly consumed foods. Mothers' dietary intake was assessed over a 24-h recall period, in which mothers were asked to report whether they had consumed foods from 18 different categories. Mother's height and weight were measured, and BMI calculated. Household surveys were used to collect data at baseline to assess agriculture production, ownership and control over assets, mothers health, nutrition and hygiene-related knowledge, household and mothers dietary diversity, and mothers perceptions of intrahousehold processes related to women's empowerment.<br><br>**Behaviour change** Women's empowerment, measured using a 30-question survey. | Significant increases in mothers' fruit intake, small increases in meat/poultry intake, and dietary diversity. Prevalence of underweight was significantly reduced. Although the changes in BMI did not differ between mothers in treatment and control villages, there was a marginally significant interaction indicating that underweight mothers had a greater increase in BMI than did mothers who were not underweight. Mothers' overall empowerment score increased: meeting with women, purchasing decisions and health care decisions. | Low 9 |

*(Continued)*

**Table 3.** (Continued)

| | Author, publication year, country | Study design and intervention | Behaviour change wheel intervention function | Setting and participants | Assessment | Outcome and Take-home message | Risk of Bias Score |
|---|---|---|---|---|---|---|---|
| 73 | Olney et al, 2015, Burkina Faso | Evaluation of 2-year RCT enhanced-homestead food production. This focuses on agricultural production and behaviour change communication by dedicating land to women's production increase production and consumption of nutrient-rich foods and to generate additional income. | Education, Training, Persuasion, Environmental restructuring, Modelling. | 55 in eastern Burkina Faso with 55 all women with children 3–13months. | **Nutrition** Household consumption of individual food groups was examined by using detailed consumption data (7-d recall) on 57 commonly consumed foods. Mothers dietary intake was assessed over a 24-h recall period, in which mothers were asked to report whether they had consumed foods from 18 different categories. Mother's height and weight were measured and BMI calculated. Household surveys were used to collect data at baseline to assess agriculture production, ownership and control over assets, mothers health, nutrition and hygiene-related knowledge, household and mothers dietary diversity, and mothers perceptions of intrahousehold processes related to women's empowerment. **Behaviour change** Women's empowerment, measured using a 30-question survey. | The intervention significantly improved several child outcomes, including wasting (marginal), diarrhoea, Hb, and anaemia, especially among the youngest children. No significant impacts on stunting or underweight prevalence. Specifically, the behaviour change communication, that enabled women's empowerment, and are likely to be transformational and to have long-lasting benefits for women and their families. | Low 9 |
| 74 | McKune et al, 2020, Burkina Faso | Cluster randomized controlled trial behaviour change intervention to increase child egg consumption in rural Burkina Faso. | Modelling, Enablement, Education, Persuasion, Training. | 260 mother-child dyads with children aged 4 to 17 months from 18 villages in Kaya Department in Burkina Faso. | **Nutrition** Anthropometric measures including WLZ, WAZ, LAZ Women's decision making about eggs. Egg consumption reported by mothers. | Intervention significantly increased egg consumption and women's decision-making about eggs. Intervention significantly decreased infant wasting and underweight. When coupled with the gift of chickens, the behaviour change intervention yielded a greater increase in egg consumption and significantly reduced wasting and underweight. | Low 7 |
| 75 | Nielsen et al, 2018, Burkina Faso | Process evaluation of 2-year RCT enhanced-homestead food production. This focuses on agricultural production and behaviour change communication by dedicating land to women's production increase production and consumption of nutrient-rich foods and to generate additional income. | Education, Training, Persuasion, Environmental restructuring, Modelling. | 55 in eastern Burkina Faso with 55 all women with children 3–13months. | **Nutrition** Household consumption of individual food groups was examined by using detailed consumption data (7-d recall) on 57 commonly consumed foods. Mothers' dietary intake was assessed over a 24-h recall period, in which mothers were asked to report whether they had consumed foods from 18 different categories. Mother's height and weight were measured, and BMI calculated. Household surveys were used to collect data at baseline to assess agriculture production, ownership and control over assets, mothers' health, nutrition and hygiene-related knowledge, household and mothers dietary diversity, and mothers perceptions of intrahousehold processes related to women's empowerment. | Gaps included input constraints, knowledge gaps among community agents in agriculture and young child nutrition practices, and lower than expected activity by community volunteers. In response, staff developed measures to overcome water constraints and expand vegetable and poultry production, retrained volunteers in certain techniques of food production and counselling for nutrition behaviour change, added small incentives to motivate volunteers, and shaped both immediate and long-term changes to the programme model. Collaboration between researchers and implementers can improve programme effectiveness, project staff capacity, and advance delivery science. | Low 6 |

*(Continued)*

**Table 3.** (*Continued*)

| | Author, publication year, country | Study design and intervention | Behaviour change wheel intervention function | Setting and participants | Assessment | Outcome and Take-home message | Risk of Bias Score |
|---|---|---|---|---|---|---|---|
| | | | | | **Behaviour change** Women's empowerment, measured using a 30-question survey. | | |
| **6 BCW IF** | | | | | | | |
| 76 | Han et al, 2021, Ethiopia | Cluster randomized control trial four-month-long BCC and food vouchers in Ethiopia. | Enablement Education, Modelling, Persuasion, Incentivisation, Training. | Mothers with at least one child aged between 4 and 20 months in Ejere district (woreda) located in the Oromia region of central Ethiopia. | **Nutrition** IYCF knowledge scores and child dietary diversity score (CDDS) Child anthropometry such as height-for-age Z scores (HAZ) and stunting as well as weight-for- height Z scores (WHZ) and wasting. Food consumption score (FCS). Weekly household food expenditure in the past seven days. | Improvements in child-feeding practices and a reduction in chronic child undernutrition only when BCC and vouchers were provided together. BCC or voucher alone had limited impacts. Importance of adding an effective educational component to existing transfer programmes | Low 7 |
| 77 | Becquey et al, 2022, Burkina Faso | Cluster-RCT of a nutrition- and gender-sensitive poultry value chain intervention (SELEVER) with and without a hygiene and sanitation component (WASH). SELEVER combined poultry revenue generation, women's empowerment, and nutrition Behaviour Change Communication (BCC). Any input or food distribution for free was specifically excluded. | Enablement, Training, Persuasion, Modelling, Environmental reconstructing, Education. | Women and children aged 2 to 4 years in Burkina Faso | Interactive 24-h recall (Individual Dietary Diversity Score (IDDS), probability of adequacy (PA) of intake for iron, zinc, and vitamin A, mean probability of adequacy (MPA) in micronutrient intake of 11 micronutrients, minimum acceptable diet in children aged 6–23 months) | Both intervention arms (SELEVER with and without WASH) increased the probability of adequacy of iron intakes in women; no further impact on primary outcomes Increased egg consumption in index children Information-only-based value chain interventions may not have meaningful positive effects on diets of women and children in the lean season in settings with largely inadequate diets | Low 6 |
| 78 | Santoso et al, 2021, Tanzania | Cluster-randomized trial nutrition-sensitive agroecology education Intervention in Tanzania. | Education, Environmental restructuring, Persuasion, Training, Incentivisation, Modelling. | One man and one woman "mentor farmer" were elected from each intervention village. 591 households from Singida Rural District, in Tanzania. Surveyed infants 2 years and younger. | **Nutrition** Dietary Diversity score, minimum meal frequency HAZs and WHZs Household Food Insecurity Access Scale. **Psychosocial** Women's Empowerment in Agriculture Index Perceived Social Support Scale. | After 2 growing seasons, the intervention improved children's dietary diversity score by 0.57 food groups. The percentage of children achieving minimum dietary diversity (≥4 food groups) increased by 9.9 percentage points during the postharvest season. The intervention significantly reduced household food insecurity. No significant impact on child anthropometry. The intervention also improved a range of sustainable agriculture, women's empowerment, and women's well-being outcomes. | Low 5 |
| 79 | Aubel et al, 2004, Senegal | Mixed method evaluation of an intervention on Participatory communication and empowerment education approach with grandmothers, including songs, stories, group discussions. | Modelling, Education, Persuasion Enablement, Training, Education. | Participants were grandmothers in 13 of the 60 rural villages in western Senegal supported by the child health programme. | Interviews (n=15) and a questionnaire (8 items) (n=134) were conducted and administered to grandmothers before and after the intervention. Mothers also filled out a questionnaire (n=100) on the advice grandmother's give about nutrition. | Grandmother's nutrition knowledge and advice to reproductive women improved 12 months post intervention. Nutrition-related practices of younger women improved as a result of grandmother's advice for both pregnancy and infant feeding. Future maternal and child health programmes should involve grandmothers and build their intrinsic commitment to family health. | High -5 |

(*Continued*)

**Table 3.** (Continued)

| Author, publication year, country | Study design and intervention | Behaviour change wheel intervention function | Setting and participants | Assessment | Outcome and Take-home message | Risk of Bias Score |
|---|---|---|---|---|---|---|
| | | | | Process evaluation included focus groups conducted 1 year later. | | |

Abbreviations include: ANC = Antenatal care, ASFs = Animal source foods, BMI = Body mass index, CPNP = Community-based Participatory Nutrition, DDS = Dietary Diversity Score, FCS = Food Consumption Score, FFQ = Food frequency questionnaire, HAZ = Height for Age Z score, Hb = Haemoglobin, HBS = Household Budget Survey, HEWs = Health extension workers, IDDS = Individual Dietary Diversity Score, IPC = Interpersonal communication, IYCF = Infant and young child feeding, KAP = Knowledge Attitude Practice, LAZ = Length for Age Z score, MAD = Minimum acceptable diet, MDD = Minimum diet diversity, MMF = Minimum meal frequency, MUAC = Mid-Upper Arm Circumference, SBCC = Social and Behaviour Change Communication, WAZ = Weight for Age Z score, WHZ = Weight for Height Z score, WLZ = Weight to Length Z score, UCT = Unconditional Cash Transfer, WHO = World Health Organisation, VAWI = Violence Against Women instrument

the description. The Strength of Recommendation Taxonomy (SORT) was used to appraise the overall quality, quantity and consistency of evidence and provide a grading for the recommendations of this review [41]. Tables 3 and 4 report the descriptions of each paper, the risk of bias score and the direction effect according to the hypothesis.

## Results

1193 papers were identified leaving 995 studies after duplicates were removed to be screened by titles and abstracts. From this screening, 158 papers were eligible to progress to full text screening. Twenty-three further articles, eight new interventions studies and 15 papers related to studies already included in this review, were found during the bibliography search and expert consultation. A total of 79 articles were included (Fig 3).

The 79 articles were published between 2004–2022 and included cluster randomised control trials (RCT)(n = 38) [20, 42–78], quantitative studies(n = 17) [79–99], study evaluations (n = 13) [55, 58, 69, 100–110], mixed methods studies(n = 5) [80, 81, 111–113], qualitative studies(n = 4) [114–117], and national case studies(n = 2) [118, 119]. Behaviour change interventions were identified from 18 sub-Saharan African countries: Ethiopia(n = 19) [45, 60, 61, 64, 66, 68, 69, 71, 84–87, 92, 96, 103–105], Malawi(n = 10) [46, 47, 72, 78, 101, 102, 106, 109, 116, 117], South Africa(n = 8) [43, 44, 48, 49, 67, 79–81], Kenya(n = 5) [42, 88, 111, 112, 114], Burundi(n = 5) [51–55], Nigeria(n = 6) [74, 82, 89, 91, 93, 115], Burkina Faso(n = 6) [51, 57, 58, 75, 77, 120], Ghana(n = 4) [83, 108, 119],Tanzania(n = 3) [50, 73, 94], Madagascar(n = 2) [20, 76], Zambia(n = 2) [99, 107], Senegal(n = 2) [95, 113] and one each from, Central African Republic [118], Mali [65], Mozambique [98], Rwanda [70], Togo [63], and Uganda [56]. One study included findings from four countries including Burkina Faso, Tanzania, Senegal, and Cote d'Ivoire [97]. A total of 30 studies were graded as low risk of bias, 38 as medium, and 11 as high (Table 2). Studies were not removed if they were deemed low quality as findings might still have been informative providing novel evidence on using behaviour change and nutrition components together in an intervention, accepting that there was some bias in the intervention design. This was accounted for in the analysis and discussion of this paper. However, all findings are presented alongside their risk of bias to highlight which studies' findings may be less credible. The studies deemed as low quality are not given as much weight in this review due to the possible bias and accuracy of results and interpretation. Studies of low quality were not included in the vote counting analysis (Table 4).

**Table 4. Effect direction plot.**

| Author, year, country | Study design | Risk of Bias | Sample size | BCW IF | Outcomes | | | |
|---|---|---|---|---|---|---|---|---|
| | | | | | Anthropometric markers | Dietary outcomes | Psychosocial outcomes | Other |
| **1 BCW IF** | | | | | | | | |
| Shapu et al, 2022, Nigeria | RCT | Low | 417 participants | Education | | ↓ | ↑ | |
| Diddana et al, 2018, Ethiopia | RCT | Low | 138 participants | Education | | ↑[1] | ↑[1] | |
| Downs et al, 2019, Senegal | Quantitative | Medium | 47 households | Persuasion | | ↔[2] | ↑[2] | |
| Kang et al, 2017a, Ethiopia | Evaluation | Medium | 2064 mother-child dyads | Enablement | | ↔ | | ↑ |
| Kang et al 2017c, Ethiopia | Evaluation | Medium | 2064 mother-child dyads | Enablement | ↔ | | | |
| Kang et al, 2017b, Ethiopia | Evaluation | Medium | 1987 participants | Enablement | | ↑ | | ↑ |
| | | | | **Proportion of effects favouring the intervention** | 0.00 | 0.40 | 1.00 | 1.00 |
| | | | | **95%-CI** | 0.00, 0.79 | 0.12, 0.77 | 0.44, 1.00 | 0.34, 1.00 |
| | | | | **p-value** | NA | 1.000 | 0.250 | 0.500 |
| **2 BCW IF** | | | | | | | | |
| Demilew et al, 2020, Ethiopia | RCT | Low | 694 participants | Enablement, Persuasion | ↑[1] | | ↑[1] | |
| Felegush et al, 2018, Ethiopia | RCT | Low | 202 participants | Environmental restructuring, Training | ↑[1] | | ↑[1] | |
| Becquey et al, 2019, Burkina Faso | RCT | Medium | 2150 participants | Enablement, Persuasion | ↑ | | | |
| Tamiru et al 2017, Ethiopia | Quantitative | Medium | 992 participants | Education. Modelling | ↑ | ↔[1] | | |
| Morris et al, 2012, Uganda | RCT | Medium | 237 participants | Education, Enablement | | | ↑ | |
| Antwi et al, 2020, Ghana | Evaluation | Medium | 351 participants | Education, Training | ↑[1] | ↓[1] | ↑[1] | |
| Mushaphi et al, 2017, South Africa | Mixed methods | Medium | 129 children and 125 caregivers | Education. Enablement | | ↑[1] | | |
| Mushaphi et al, 2015, South Africa | Mixed methods | Medium | 129 children and 125 caregivers | Education. Enablement | ↑[1] | NA[3] | | |
| | | | | **Proportion of effects favouring the intervention** | 1.00 | 0.25 | 1.00 | NA |
| | | | | **95%-CI** | 0.61, 1.00 | 0.05, 0.70 | 0.51, 1.00 | NA |
| | | | | **p-value** | **0.031*** | 0.625 | 0.125 | NA |
| **3 BCW IF** | | | | | | | | |
| Jemmott et al, 2011, South Africa | RCT | Low | 1057 participants | Enablement, Training, Education | | ↑ | ↑ | ↑ |
| Jemmott et al, 2019, South Africa | RCT | Low | 1057 participants | Enablement, Training, Education | | ↑ | ↑ | ↑ |
| Galasso et al, 2019, Madagascar | RCT | Low | 3738 households | Training, Education, Enablement | ↔ | | ↓ | |
| Abiyu et al, 2020, Ethiopia | RCT | Low | 612 participants | Training, Modelling, Enablement | | ↔ | | |
| Walton et al, 2017, Kenya | Quantitative | Low | 111 participants | Modelling, Education, Persuasion | | ↑[1] | ↑[1] | |

*(Continued)*

**Table 4.** (Continued)

| Author, year, country | Study design | Risk of Bias | Sample size | BCW IF | Outcomes | | | |
|---|---|---|---|---|---|---|---|---|
| | | | | | Anthropometric markers | Dietary outcomes | Psychosocial outcomes | Other |
| Passarelli et al, 2020, Ethiopia | RCT | Low | 829 participants | Environmental restructuring, Enablement, Persuasion | ↑ | ↑ | ↑ | ↑ |
| Leroy et al, 2019, Burundi | RCT | Low | 11906 observations | Enablement, Education, Persuasion | | ↑ | | |
| Huybregts et al 2019, Mali | RCT | Low | 1132 participants | Enablement, Persuasion, Incentivisation | ↑[1] | | | |
| Olney et al, 2019, Burundi | RCT | Low | 1323 participants | Enablement, Education, Persuasion | ↑ | ↔ | ↑ | ↑ |
| Leroy et al, 2018, Burundi | RCT | Low | ~ 3550 participants | Enablement, Education, Persuasion | ↑ | | | |
| Leroy et al, 2016, Burundi | RCT | Low | 2533 participants | Enablement, Education, Persuasion | | ↑ | | ↑ |
| Mulualem et al, 2016, Ethiopia | RCT | Low | 160 mother-child dyads | Education, Training, Incentivisation | ↑[1] | | ↑[1] | |
| Ogunsile & Ogundele, 2016, Nigeria | Quantitative | Medium | 143 participants | Environmental restructuring, Enablement, Training | | | ↑ | |
| Tariku et al 2015, Ethiopia | Quantitative | Medium | 166 mother-infant dyads | Education, Training, Incentivisation | | ↑[1] | ↑[1] | |
| Galasso et al, 2009, Madagascar | RCT | Medium | 18177 participants | Training, Education, Enablement | ↑ | | | |
| Saaka et al, 2021, Ghana | Quantitative | Medium | 712 participants | Persuasion, Modelling, Education | ↓ | ↑ | ↑ | |
| Yetnayet et al 2017, Ethiopia | Quantitative | Medium | 200 participants | Enablement, Training, Education | | ↑[1] | ↑[1] | |
| Hurley et al, 2021, Malawi | Evaluation | Medium | 2404 participants | Enablement, Persuasion, Education | ↑ | ↑ | ↔ | ↑ |
| De Villiers et al, 2016, South Africa | RCT | Medium | 998 participants | Training, Modelling, Education | | | ↔[1] | |
| Steyn et al, 2015, South Africa | RCT | Medium | 998 participants | Training, Modelling, Education | | ↔ | | |
| Grant et al, 2022, Tanzania | Quantitative | Medium | 547 participants | Education, Enablement, Persuasion | | ↑ | ↑ | |
| Mutiso et al, 2018, Kenya | Quantitative | Medium | 665 participants | Education, Persuasion, Enablement | | ↑ | | |
| Workicho et al, 2021, Ethiopia | Quantitative | Medium | 508 participants | Modelling, Persuasion, Education | | | ↑ | |
| Skar et al, 2019, Mozambique | Quantitative | Medium | 350 participants | Training, Enablement, Modelling | | | ↑ | |
| | | | **Proportion of effects favouring the intervention** | | 0.78 | 0.80 | 0.81 | 1.00 |
| | | | **95%-CI** | | 0.45, 0.94 | 0.55, 0.93 | 0.57, 0.93 | 0.61, 1.00 |
| | | | **p-value** | | 0.180 | **0.035**[*] | **0.021**[*] | **0.031**[*] |
| **4 BCW IF** | | | | | | | | |
| Kim et al, 2019a, Ethiopia | Evaluation | Low | 2720 participants | Persuasion, Environmental restructuring, Modelling, Enablement | ↑ | ↑ | | |
| Byrd et al, 2019, Kenya | RCT | Low | 4987 participants | Enablement, Training, Education, Persuasion | | ↑[1] | | |

*(Continued)*

**Table 4.** (Continued)

| Author, year, country | Study design | Risk of Bias | Sample size | BCW IF | Outcomes | | | |
|---|---|---|---|---|---|---|---|---|
| | | | | | Anthropometric markers | Dietary outcomes | Psychosocial outcomes | Other |
| Mlinda et al, 2018, Tanzania | RCT | Low | 110 participants | Education, Training, Enablement, Environmental restructuring | | | ↑ | |
| Kumar et al, 2018, Zambia | RCT | Low | 2346 participants | Modelling, Education, Enablement, Persuasion | ↑ | ↑ | | ↑ |
| Rosenberg et al, 2018, Zambia | Quantitative | Low | 3044 households | Modelling, Education, Enablement, Persuasion | | ↑ | | ↓ |
| Flax et al, 2021, Rwanda | RCT | Medium | 462 participants | Enablement, Education, Training, Persuasion | | ↓ | | |
| Katenga-Kaunda et al, 2021, Malawi | RCT | Medium | 195 participants | Modelling, Enablement, Education, Persuasion | | ↑ | ↑ | |
| Briaux et al, 2020, Togo | RCT | Medium | 2658 participants | Persuasion, Incentivisation, Enablement, Training | | ↑ | ↔ | ↑ |
| Lion et al, 2018, Nigeria | RCT | Medium | 1054 participants | Enablement, Education, Persuasion, Incentivisation | | ↑[1] | | |
| Katenga-Kaunda et al, 2020, Malawi | RCT | Medium | 257 participants | Modelling, Enablement, Education, Persuasion | | ↑ | ↑ | |
| Kim et al, 2016, Ethiopia | Quantitative | Medium | 2956 participants | Persuasion, Environmental restructuring, Modelling, Enablement | ↔ | ↑ | | |
| Flax et al, 2022, Nigeria | Quantitative | Medium | 992 participants | Enablement, Education, Training, Persuasion | | ↑[2] | ↔[2] | |
| Jacobs et al 2013, South Africa | Quantitative | Medium | 325 participants | Enablement, Education Training, Persuasion | | | ↔[1] | |
| Kim et al, 2019b, Ethiopia | RCT | Medium | 1720 participants | Persuasion, Environmental restructuring, Modelling, Enablement | | ↑ | | |
| Nordhagen and Klemm, 2018, Burkina | Quantitative | Medium | 1110 households | Training, Environmental restructuring, Persuasion, Modelling | | ↑[1] | | |
| | | | | **Proportion of effects favouring the intervention** | 0.67 | 0.92 | 0.50 | 0.67 |
| | | | | **95%-CI** | 0.21, 0.94 | 0.67, 0.99 | 0.19, 0.81 | 0.21, 0.94 |
| | | | | **p-value** | 1.000 | **0.003**[*] | 1.000 | 1.000 |
| **5 BCW IF** | | | | | | | | |
| Gelli et al, 2020, Malawi | RCT | Low | 1248 participants | Enablement Education, Modelling, Persuasion, Environmental restructuring | | ↑ | | |
| Gelli et al, 2018, Malawi | RCT | Low | 1122 participants | Enablement Education, Modelling, Persuasion, Environmental restructuring | | ↑ | | |
| Olney et al, 2016, Burkina Faso | RCT | Low | 1767 participants | Education, Training, Persuasion, Environmental restructuring, Modelling | ↔ | ↔ | ↑ | |
| Olney et al, 2015, Burkina Faso | RCT | Low | 55 villages | Education, Training, Persuasion, Environmental restructuring, Modelling | ↑[1] | ↑[1] | | ↑[1] |
| McKune et al, 2020, Burkina Faso | RCT | Low | 260 mother-child dyads | Modelling, Enablement, Education, Persuasion, Training | ↑ | ↑ | ↑ | |

*(Continued)*

**Table 4.** (Continued)

| Author, year, country | Study design | Risk of Bias | Sample size | BCW IF | Outcomes | | | |
|---|---|---|---|---|---|---|---|---|
| | | | | | Anthropometric markers | Dietary outcomes | Psychosocial outcomes | Other |
| Nielsen et al, 2018, Burkina Faso | RCT | Low | 260 mother-child dyads | Education, Training, Persuasion, Environmental restructuring, Modelling | ↑ | ↑ | ↑ | |
| | | | | Proportion of effects favouring the intervention | 0.75 | 0.83 | 1.00 | 1.00 |
| | | | | 95%-CI | 0.30, 0.95 | 0.44, 0.97 | 0.44, 1.00 | 0.21, 1.00 |
| | | | | p-value | 0.625 | 0.219 | 0.250 | 1.000 |
| **6 BCW IF** | | | | | | | | |
| Han et al, 2021, Ethiopia | RCT | Low | 290 mother-child dyads | Enablement Education, Modelling, Persuasion, Incentivisation, Training | ↑ | ↑ | | |
| Becquey et al, 2022, Burkina Faso | RCT | Low | 1054 households | Enablement, Training, Persuasion, Modelling, Environmental reconstructing, Education | | ↔ | | |
| Santoso et al, 2021, Tanzania | RCT | Low | 591 households | Education, Environmental restructuring, Persuasion, Training, Incentivisation, Modelling | ↔ | ↑ | ↑ | |
| | | | | Proportion of effects favouring the intervention | 0.50 | 0.67 | 1.00 | NA |
| | | | | 95%-CI | 0.095, 0.91 | 0.21, 0.94 | 0.21, 1.00 | NA |
| | | | | p-value | 1.000 | 1.000 | 1.000 | NA |
| **OVERALL** | | | | | | | | |
| | | | | Proportion of effects favouring the intervention | 0.76 | 0.74 | 0.82 | 0.92 |
| | | | | 95%-CI | 0.57, 0.89 | 0.60, 0.84 | 0.66, 0.91 | 0.65, 0.99 |
| | | | | p-value | **0.015*** | **0.002*** | **<0.001*** | **0.006*** |

Footnote:

* p value < 0.05, [1] result not considered due to study authors not conducting a comparison because of low and unequal numbers in the intervention and control group;
Abbreviations: RCT = Randomised controlled trial, 95%-CI = 95% confidence interval (Wilson score interval)

Descriptions of intervention design, outcomes and implications are presented in Table 3. The results are summarised below by the impact the interventions had on 1) anthropometric markers, 2) dietary outcomes, 3) psychosocial outcomes and 4) behavioural intervention components. Using SORT to grade the recommendations, we conclude that the use of behaviour change components within nutrition interventions improves maternal and child nutrition and psychosocial outcomes. Our recommendation is graded as SORT B due to: the use of a variety of study designs including RCTs, evaluations, quantitative and qualitative studies; a range of low-high risk of bias; and the direction of effect of studies to meet the hypothesis [41].

## Anthropometric markers

Overall, there was evidence showing that behaviour change nutrition interventions have an effect on anthropometric markers such as height and weight. Nineteen out of the 25 studies

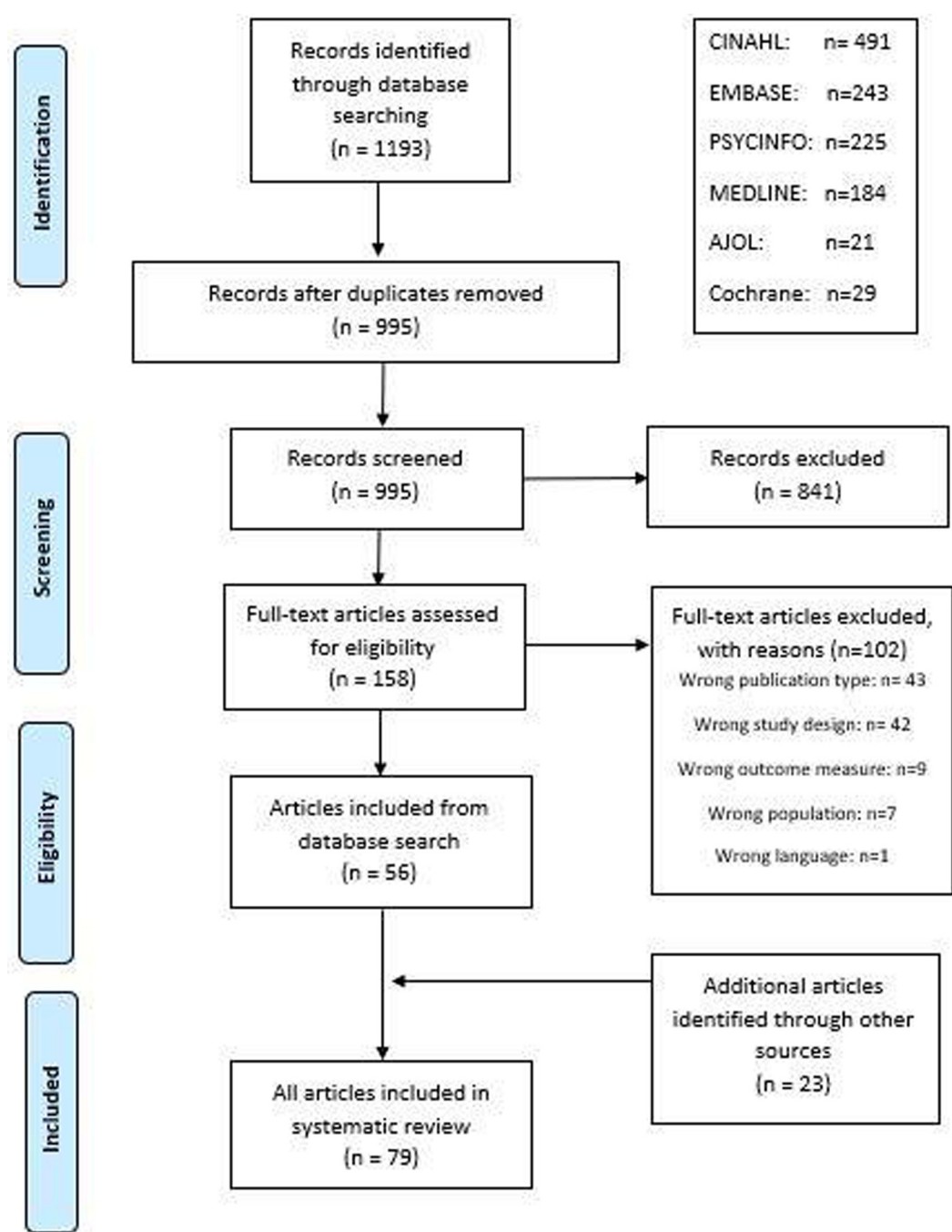

**Fig 3. Flowchart of study selection PRISMA diagram.**

favoured the intervention (76% [57%,89%], p = 0.015); 16 out of the 25 studies were of low risk of bias. Out of these, 13 favoured the intervention.

A mother and child lipid supplementation RCT combined with maternal nutrition counselling in Madagascar reported an increase in infants' length-for-age z-scores (LAZ) of 0.210–0.216SD and stunting reduced by 8.2–9.0% compared to the control (RoB = Medium, DE = ↑ [76]; RoB = Low, DE = ↔ [20]). Similar interventions in Malawi found mixed results, where

one intervention had no effect on infant LAZ scores or stunting prevalence, but both interventions found significantly higher weight-for-length z-score (WLZ) and mid-upper arm circumference at 18 months to three years follow-up (RoB = Medium, DE = N/A [102]; RoB = High, DE = N/A) [101]; RoB = High, DE = N/A) [106]; RoB = Medium, DE = ↑ [109]). These results have to be viewed with caution, however, since the trial was low quality being not randomised or blinded. Interventions that provided food rations or vouchers and behaviour change communication to mothers in Burundi and Ethiopia, showed reduced prevalence of child stunting in the intervention compared to the control arm (RoB = Low, DE = ↑ [51, 53, 71]).

An agricultural production and behaviour change communication intervention for mothers in Burkina Faso significantly improved child wasting, diarrhoea, haemoglobin levels and rates of anaemia, especially among the youngest, as well as reducing underweight prevalence (RoB = Low, DE = ↔ [57]; RoB = Low, DE = ↑ [58]). One chicken production intervention improved infant HAZ and weight-for-age z-scores (WAZ) both with and without behaviour change communication in Ethiopia (RoB = Low, DE = ↑ [66]). An agricultural (tools and agricultural extension workers), behaviour change communication and nutrition counselling intervention in Ethiopia reduced prevalence of stunting (36.3% to 22.8%) in children aged 6–23 months(RoB = Low, DE = ↑ [69];RoB = Medium, DE = ↔) [100]. One RCT that trained local Tanzanian women and farmers to deliver agricultural education to households as well as them receiving seedlings did not find significantly improved infant HAZs and WHZs scores RoB = Low, DE = ↔) [73].

## Dietary outcomes

There was evidence that behaviour change nutrition interventions influence dietary outcomes, with 34 out of 46 studies favouring the intervention (74% [60%, 84%], p = 0.002). Twenty-three studies were judged to be at a low risk of bias, and 18 of these favoured the intervention.

Two behaviour change communication interventions increased maternal dietary diversity in Ethiopia(RoB = Medium, DE = ↔ [103]; RoB = Medium, DE = ↑) [104]) and in Burundi (RoB = Low, DE = ↔ [51]; RoB = Low, DE = ↑ [52]). One behaviour change communication RCT found increased infant consumption of milk, but this was not significant (RoB = Medium, DE = ↓ [70]). Two interventions based on nutrition education found small yet insignificant changes in the nutrient intake of children in Malawi (RoB = Low, DE = ↑ [46, 47]) and South Africa(RoB = Medium, DE = ↑ [80]; RoB = Medium, DE = N/A [81]). Yet one RCT found significant infant dietary diversity improvement with a combined nutrition education and counselling approach in Malawi (RoB = Medium, DE = ↑ [72, 78]). One intervention underpinned by Social Cognitive Theory and the Theory of Planned Behaviour had school children in South Africa meeting 5-a-day fruit and vegetable guidelines (RoB = Low, DE = ↑ [48, 49]).

Two psychosocial interventions increased pregnant women's and mothers' dietary diversity, and household food consumption in the Central African Republic (RoB = High, DE = N/A [118]) and Ethiopia (RoB = Medium, DE = ↑ [104]; RoB = Medium, DE = ↔ [103]). A intervention that delivered radio dramas on maternal nutrition in Ghana found improved infant dietary diversity (RoB = High, DE = N/A [110]). One RCT that trained local Tanzanian women and farmers to deliver agricultural education to households as well as them receiving seedlings found significantly improved infant dietary diversity and food insecurity (RoB = Low, DE = ↑ [73]).

## Psychosocial outcomes

We found evidence that interventions including behaviour change functions have an effect on psychosocial outcomes, with 27 out of 33 studies favouring the intervention (82% [66%, 91%], p < 0.001). Sixteen studies had a low risk of bias, of which 15 favoured the intervention.

Interventions aimed to address psychosocial outcomes including nutrition knowledge (n = 17) [43–45, 47–49, 60, 63, 72, 74, 82, 86, 88, 89, 108, 113, 114], practices (e.g. food/recipes preparation)(n = 5) [45, 82, 86, 88, 90], attitudes (n = 6) [45, 48, 49, 64, 82, 86], intentions (n = 4) [48, 49, 64, 91] and confidence in eating healthy food (n = 1) [114]. These were variously underpinned by behaviour change theory(n = 10) [45, 48, 49, 60, 64, 82, 86, 88, 90, 108], strategies to increase self-efficacy (n = 5) [43–45, 79, 91], behaviour change communication (n = 2) [47, 63] or behaviour change techniques (n = 1) [89].

In two interventions women's capability to feed themselves and their children more nutritious food increased, including interventions based on behaviour change communication in Burkina Faso (RoB = Low, DE = ↑ [57, 58]) and Zambia (RoB = Low, DE = ↑ [107]), and a psychosocial intervention in Ghana (RoB = High, DE = N/A [83]). Five psychosocial interventions achieved positive psychosocial outcomes, which improved caregiver feeding behaviours. This included increasing women's psychological well-being in the Central African Republic (RoB = High, DE = N/A [118]) and Uganda (RoB = Medium, DE = ↑ [56]). Caregivers of children with cerebral palsy in Tanzania improved their feeding skills (RoB = Low, DE = ↑ [50]). Caregiver relationships and communication improved leading to children's prosocial behaviour in Mozambique (RoB = Medium, DE = ↑ [98]). An intervention which delivered radio dramas on maternal nutrition in Ghana found increased mother's knowledge of feeding practices (RoB = High, DE = N/A [110]). Qualitative evaluations identified barriers to the success of behavioural interventions including availability of, and access to, nutritious and affordable foods and less time for mothers to care for children (RoB = High, DE = N/A [116]).

## Behavioural intervention components

Interventions were coded on the basis of involving one or more Behaviour Change Wheel intervention functions (Fig 1) [28]. Functions (S3 Table) included education(n = 56) [20, 42–44, 47–50, 52, 54–57, 58, 60, 70–81, 83, 84, 87, 89–94, 103–105, 107–112, 115–121], enablement(n = 56) [20, 42, 47–50, 52, 54–57, 58, 61–72, 75–82, 85, 86, 89–91, 93, 94, 96, 98–107, 109, 113–115, 117, 119–121], persuasion(n = 46) [42, 52, 54, 55, 57, 58, 62–66, 68–73, 75, 77, 78, 88–90, 92–97, 99–102, 106, 107, 110, 113, 117, 119–121], training(n = 39) [20, 42–45, 48–50, 57, 58, 61, 63, 67, 70, 71, 73, 75–77, 79, 82–86, 91, 93, 97–99, 101, 102, 106, 108, 111–115, 120], modelling(n = 39) [43, 44, 48, 49, 57, 58, 61, 67–69, 71–73, 75, 77, 78, 87, 92, 96–98, 100, 110–112, 117, 120], environmental restructuring (n = 17) [45, 47, 50, 57, 58, 66, 68, 69, 73, 77, 82, 96, 97, 99, 100, 116, 120], incentivisation (n = 11) [63, 65, 71, 73, 84, 85, 89, 111, 116, 118]. There were no interventions that included restriction or coercion. Table 5 indicates that in interventions where the Behaviour Change Wheel intervention functions persuasion, environmental restructuring and incentivisation were used, body composition and diet intake improved.

Education, enablement and training appeared to have less impact on these maternal and infant body composition and diet intake outcomes. However, psychosocial outcomes seemed to improve in interventions with training, incentivisation, modelling and education functions, and less so in interventions which included enablement such as nutrient supplementation. Interventions with multiple Behaviour Change Wheel intervention functions tended also to be those that improved maternal and child nutrition and health outcomes. The optimal number of intervention functions is not possible to determine because so few of the interventions had more than three. Our ability to draw conclusions about the effectiveness of intervention functions is limited by the fact that many of the studies do not measure all outcomes; measurements of nutritional intake are often omitted.

Table 6 describes the way in which people's capability, opportunity and motivation are addressed in interventions to improve maternal and child nutrition and health outcomes. For

**Table 5. The numbers of interventions using Behaviour Change Wheel intervention functions and the percentage of those where outcomes improved, did not improve or were not measured.**

| Behaviour Change Wheel Intervention Function (BCW IF) | Number of interventions including each function | Anthropometric markers | | | Dietary outcomes | | | Psychosocial outcomes | | |
|---|---|---|---|---|---|---|---|---|---|---|
| | | Significant | Non-Significant | Not measured | Significant | Non-Significant | Not measured | Significant | Non-Significant | Not measured |
| Enablement | 56 | 30.4% | 7.1% | 67.9% | 51.8% | 12.5% | 37.5% | 51.8% | 5.4% | 44.6% |
| Education | 56 | 17.9% | 10.7% | 71.4% | 53.6% | 14.3% | 32.1% | 73.2% | 3.6% | 23.2% |
| Persuasion | 46 | 34.2% | 15.8% | 52.6% | 57.9% | 10.5% | 34.2% | 52.6% | 5.3% | 44.7% |
| Training | 39 | 25.6% | 10.3% | 64.1% | 43.6% | 17.9% | 38.5% | 76.9% | 7.7% | 15.4% |
| Modelling | 39 | 20.5% | 12.8% | 69.2% | 64.1% | 12.8% | 25.6% | 64.1% | 2.6% | 35.9% |
| Environmental reconstructing | 17 | 29.4% | 17.6% | 58.8% | 70.6% | 5.9% | 29.4% | 58.8% | 5.9% | 41.2% |
| Incentivisation | 11 | 36.4% | 9.1% | 54.5% | 72.7% | 0.0% | 27.3% | 90.9% | 0.0% | 9.1% |
| 1 BCW IF | 6 | 16.7% | 0.0% | 83.3% | 66.7% | 16.7% | 16.7% | 33.3% | 16.7% | 50.0% |
| 2 BCW IF | 11 | 18.2% | 18.2% | 63.6% | 54.5% | 9.1% | 36.4% | 54.5% | 0.0% | 45.5% |
| 3 BCW IF | 33 | 24.2% | 9.1% | 66.7% | 36.4% | 18.2% | 45.5% | 66.7% | 0.0% | 33.3% |
| 4 BCW IF | 19 | 15.8% | 5.3% | 84.2% | 73.7% | 10.5% | 21.1% | 47.4% | 15.8% | 42.1% |
| 5 BCW IF | 6 | 50.0% | 16.7% | 33.3% | 83.3% | 0.0% | 16.7% | 100.0% | 0.0% | 0.0% |
| 6 BCW IF | 4 | 25.0% | 25.0% | 50.0% | 75.0% | 25.0% | 0.0% | 50.0% | 0.0% | 50.0% |

example, food demonstrations and provision of agricultural land address physical capability and opportunity components respectively as specified by the COM-B model as necessary to change behaviour.

## Discussion

This review of 79 studies addressed the question of whether nutrition-specific and nutrition-sensitive interventions were effective in improving maternal and child nutrition outcomes in

**Table 6. The categories of studies' intervention components analysed to the COM-B model.**

| COM-B model components | Nutrition-specific intervention components | Nutrition-sensitive intervention components | Behaviour change intervention components |
|---|---|---|---|
| **Physical Capability (e.g. skills)** | Food demonstrations Breastfeeding skills sessions | Employment skills Agricultural technique development | Mastery to increase self-efficacy Modelling |
| **Psychological Capability (e.g. knowledge)** | Nutrition education Breastfeeding education | Water, Sanitation and Hygiene (WASH) knowledge | Observational learning |
| **Physical Opportunity (e.g. environment)** | Providing food rations Supplementation and fortification Cooking utensils | Provision of agriculture land and equipment WASH facilities Unconditional cash transfers | Environmental restructuring through cues, reminders, access |
| **Social Opportunity (e.g. social network)** | Nutrition support from elders, community health workers, peers | Community support groups Gamification Women's empowerment | Community groups problem-solving Modelling Support networks |
| **Automatic Motivation (e.g. habits)** | N/A | N/A | Behaviour change communication |
| **Reflective Motivation (e.g. decision-making)** | N/A | N/A | Counselling, Goal-setting |

sub-Saharan Africa if they included behaviour change functions. It was found that those that were based on behaviour change theory, counselling and communication, produced the most significant positive impacts on infant and children nutrition outcomes. This included reduced prevalence of wasting, underweight and stunting, and improved diet outcomes including dietary diversity and total food consumption. There is little evidence that these interventions improved maternal and infant nutrient intake. This could be because changes in nutrient intake were not often measured due to a lack of comprehensive data on the nutrient composition of foods. These interventions reviewed also identified improvements in maternal psychosocial outcomes in relation to nutrition, including knowledge, practice, attitude, intention, confidence, capability and wellbeing.

Interventions which included the Behaviour Change Wheel intervention functions, i.e. incentivisation, persuasion and environmental restructuring, tended to be those that improved maternal and child nutrition and psychosocial outcomes. The COM-B model suggests that not addressing participant's motivation reduces the effectiveness of behaviour change interventions. This analysis in Table 6 demonstrates that nutrition-specific and nutrition-sensitive interventions cannot alone address motivation and require additional behaviour change functions in order to address behaviours that improve maternal and child nutrition and health outcomes.

## Strengths and limitations

This review is strengthened by the adherence to the PRISMA, Cochrane and SWiM guidelines throughout. Another strength of this systematic review is the inclusion of a wide variety of studies and using a range of intervention methods and outcomes. This provided a broad overview of the way in which behavioural functions may increase effectiveness of nutrition interventions to improve maternal and child nutrition outcomes in sub-Saharan Africa. A limitation is the potential publication bias where studies mainly report on significant results which may not provide the full picture of what interventions work and which do not. One limitation was that breastfeeding interventions were excluded. This was due to previous systematic reviews finding limited evidence of behaviour change components in the interventions [33, 34]. One weakness of the review is that mainly papers written and published in English were reviewed. Another is that not all authors of these papers explicitly identified behaviour change functions in their descriptions of interventions, meaning that we may have missed other relevant papers. All titles, abstracts and full texts were double-screened against the inclusion and exclusion criteria, however, in order to minimise this possibility. Consistency in the types of data extracted and in judgements of the quality of the studies was ensured by having two independent reviewers review the included studies.

Compared to meta-analysis the synthesis method used provides more limited information for decision making. However, it is superior to only describing studies narratively. Vote counting based on direction of effect only answers the question "is there any evidence of an effect?"[39]. This method does not provide information on the magnitude of an effect and does not account for differences in the sample sizes of the included studies. Furthermore, the uncertainty of the results for the individual categories within the effect direction plot is high due to the small number of effects contributing to the analysis.

## Implications

Improving nutritional outcomes requires behaviour change. To change nutrition behaviour, the COM-B model suggests that capability, opportunity and motivation should all be addressed [28]. Our analysis demonstrates that most nutrition-specific and nutrition-sensitive interventions only address participants' capability and opportunities. Nutrition interventions

therefore require the addition of behaviour change functions such as behaviour change communication and counselling to address participants' automatic and reflective motivation. Automatic motivation is defined as the unconscious process that "energizes and directs behaviour" [28] (p4), including habits and emotional responses. Reflective motivation describes conscious processes such as analytical decision-making.

For example, one trial in Burkina Faso gave women land and seedlings (physical opportunity) and also had volunteers facilitate women's groups using behaviour change communication that was empowering and motivating (automatic and reflective motivation [28]). Education functions of the communications in the trial in Burkina Faso may have changed the conscious decisions women were making about how to feed themselves and their children (reflective motivation) [57, 58]. The groups may have also created new social norms that encouraged women to change their dietary habits (automatic motivation).

This review linked three Behaviour Change Wheel intervention functions to improved nutrition and psychosocial outcomes. These were incentivisation, persuasion and environmental restructuring. All are hypothesised to change motivations [28]. Providing incentives, which include cash transfers and food vouchers, has been found to improve health outcomes. This could be because it increases household disposable income allowing families to spend more money on food [63, 118]. This is particularly significant in improving outcomes in the poorest communities. Another incentive used was nutritional supplements, which increased intervention attendance and uptake [65]. In contrast, failing to incentivise community health workers was given as a reason for their lack of motivation for delivering interventions [62]. Overall, there is compelling evidence that incentives can be a powerful motivating force in behaviour change interventions [122].

Persuasive communications were a common feature of many of the interventions in this review. These tended to focus on improving diet quality and diversity and were delivered through singing groups, storytelling and women's groups. The Behaviour Change Wheel indicates that persuasion, because it is often emotive and induced positive or negative feelings, engages both reflective and automatic motivations [28]. This distinguishes it from purely educational communications which work by informing conscious decision-making (reflective motivation).

In one intervention that included persuasive communication, grandmothers were encouraged through nutrition education to promote improved nutritional practices related to pregnancy and infant feeding within their families [113]. This was delivered through songs, storytelling and group discussions and made an emotional appeal to grandmothers' intrinsic commitment to family well-being. These persuasive communications were therefore explicitly designed to induce an emotional response (automatic motivation) as well as educate and inform (reflective motivation).

Environmental restructuring is the process of changing the physical and social environment, which is hypothesised to indirectly change motivation by cuing behaviour in a way which is often unconscious (automatic motivation). In the context of this review, environmental restructuring referred to provision of land, seedlings, livestock and equipment with the intention of increasing the effectiveness of community food production for both consumption and sale [57, 58, 66, 116]. It can be speculated that this provision addresses automatic motivations by increasing availability and diversity of food in these communities and supporting creation of new eating habits intended to improve diet quality.

Fig 4 illustrates the way in which all these intervention functions can be drawn together. Encouragingly, the addition of behaviour change functions to nutrition interventions may provide good value for money. An economic analysis within an intervention in Burundi calculated the relative cost of behaviour change functions accounted for approximately 13% of the

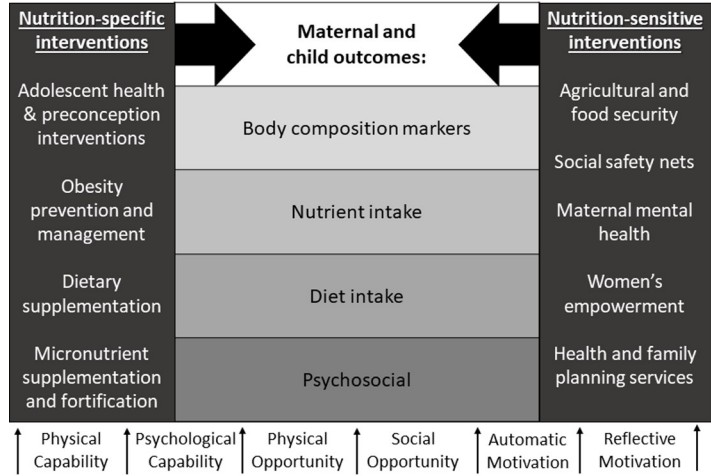

**Fig 4. Framework describing the way in which nutrition-specific and nutrition-sensitive interventions may be underpinned by behavioural concepts to improve maternal and child nutritional and psychosocial outcomes in sub-Saharan Africa.** (Adaption of Lassi et al, 2017 framework [124]).

budget whereas provision of food rations took up 30% [55]. Longitudinal studies are required to assess the impact of nutritional outcomes across children's lifecourse and as communities undergo economic, nutritional, and societal transition [123].

## Conclusion

This review indicates that nutrition-specific interventions, such as increasing access to food [20, 76], and nutrition-sensitive interventions, such as offering cash transfers [63] are not always enough to improve nutritional status of women and children in sub-Saharan Africa. Behaviour change communication and counselling are also unlikely on their own to be effective because they do not address food insecurity or nutritional deficiencies in high poverty contexts. Our findings indicate that interventions comprising all three Behaviour Change Wheel intervention functions (incentives, persuasion and environmental restructuring) were most likely to be effective in improving nutritional and psychosocial outcomes. By taking a health psychology perspective on this review, specifically drawing on concepts of automatic and reflective motivation from the Behaviour Change Wheel and COM-B model, motivation needs to be considered in designing an intervention to improve nutritional behaviour in the context of sub-Saharan Africa have been identified. To enhance the designs of these interventions, and ultimately improve the nutritional and psychosocial outcomes for mothers and infants in this region, multidisciplinary collaborations are required. Our recommendation would be to task behaviour change and nutrition experts such as health psychologists and nutritionists, as well as intervention designers, policy makers, and commissioners of services to fund and roll out these multicomponent behaviour change interventions.

## Supporting information

**S1 Table. Behaviour change systematic review search strategy.**
(DOCX)

**S2 Table. Behaviour change systematic review screening document.**
(DOCX)

**S3 Table. Behaviour Change Wheel (Michie et al, 2011).**
(DOCX)

**S4 Table. Behaviour change systematic review quality assessment scoring rubric.**
(DOCX)

**S5 Table. Behaviour change systematic review quality assessment form.**
(DOCX)

**S1 Checklist. PRISMA checklist.**
(PDF)

**S2 Checklist. SWIM checklist.**
(PDF)

## Acknowledgments

We would also like to thank the INPreP group for their contributions to this work: Engelbert A. Nonterah, Abraham Oduro, Cornelius Debpuur, James Adoctor, Paul Welaga, Edith Dambayi, Esmond W. Nonterah, Winfred Ofosu, Doreen Ayibisah, Maxwell Dalaba, Samuel Chatio (Navrongo Health Research Centre); Hermann Sorgho, Palwendé R. Boua, Adelaïde Compaoré, Kadija Ouedraogo, Karim Derra, Aminata Welgo, Halidou Tinto (Clinical Research Unit of Nanoro); Karen J. Hofman, Susan Goldstein, Agnes Erzse, Aviva Tugendhaft, Winfreda Mdewa, Ijeoma Edoka (SAMRC Centre for Health Economics and Decision Science, PRICELESS); Mark Hanson, Marie-Louise Newell, Keith M. Godfrey, Caroline Fall, Polly Hardy-Johnson (Faculty of Medicine, University of Southampton); Shane Norris, Emmanuel Cohen, Stephanie Wrottesley (SAMRC Developmental Pathways for Health Research Unit). We would also like to thank Paula Sands at the University of Southampton Health Sciences library for her support in developing the search strategy.

## Author Contributions

**Conceptualization:** Daniella Watson, Sarah H Kehoe, Kate A Ward, Mary Barker, Wendy Lawrence.

**Data curation:** Daniella Watson.

**Formal analysis:** Daniella Watson, Patience Mushamiri, Paula Beeri, Toussaint Rouamba, Sarah Jenner, Simone Proebstl.

**Investigation:** Daniella Watson, Paula Beeri, Toussaint Rouamba.

**Methodology:** Daniella Watson, Sarah Jenner, Simone Proebstl, Mary Barker, Wendy Lawrence.

**Project administration:** Daniella Watson.

**Resources:** Daniella Watson.

**Supervision:** Daniella Watson, Mary Barker, Wendy Lawrence.

**Validation:** Daniella Watson, Patience Mushamiri.

**Visualization:** Daniella Watson.

**Writing – original draft:** Daniella Watson.

**Writing – review & editing:** Daniella Watson, Patience Mushamiri, Paula Beeri, Toussaint Rouamba, Sarah Jenner, Simone Proebstl, Sarah H Kehoe, Kate A Ward, Mary Barker, Wendy Lawrence.

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
