## [Decision Letter · Decision Letter 0]

7 Jul 2022

PGPH-D-22-00527

Behaviour change interventions improve maternal and child nutrition in sub-Saharan Africa: a systematic review

Dear Dr. Watson,

Thank you for submitting your manuscript to PLOS Global Public Health. After careful consideration, we feel that it has merit but does not fully meet PLOS Global Public Health’s publication criteria as it currently stands. Therefore, we invite you to submit a revised version of the manuscript that addresses the points raised during the review process.

We look forward to receiving your revised manuscript.

Kind regards,

Roopa Shivashankar, MD, MSc

Academic Editor

Journal Requirements:

1.  Please amend your detailed online Financial Disclosure statement. This is published with the article. It must therefore be completed in full sentences and contain the exact wording you wish to be published.

2. Please update your online Competing Interests statement. If you have no competing interests to declare, please state: “The authors have declared that no competing interests exist.”

3. In the online submission form, you indicated that “Secondary data available on request”. All PLOS journals now require all data underlying the findings described in their manuscript to be freely available to other researchers, either 1. In a public repository, 2. Within the manuscript itself, or 3. Uploaded as supplementary information.

4. Please provide separate figure files in .tif or .eps format and ensure that all files are under our size limit of 10MB.

5. Please correct your figure in-text citation on page 68. 

6. We notice that your supplementary materials (appendices) are included in the manuscript file. Please remove them and upload them with the file type 'Supporting Information'. Please ensure that each Supporting Information file has a legend listed in the manuscript after the references list.

Additional Editor Comments (if provided):

Reviewers' comments:

Reviewer's Responses to Questions

**Comments to the Author**

1. Does this manuscript meet PLOS Global Public Health’s publication criteria? Is the manuscript technically sound, and do the data support the conclusions? The manuscript must describe methodologically and ethically rigorous research with conclusions that are appropriately drawn based on the data presented.

Reviewer #1: Yes

Reviewer #2: Yes

Reviewer #3: Partly

2. Has the statistical analysis been performed appropriately and rigorously?

Reviewer #1: N/A

Reviewer #2: Yes

Reviewer #3: N/A

3. Have the authors made all data underlying the findings in their manuscript fully available (please refer to the Data Availability Statement at the start of the manuscript PDF file)?

Reviewer #1: Yes

Reviewer #2: Yes

Reviewer #3: Yes

4. Is the manuscript presented in an intelligible fashion and written in standard English?

Reviewer #1: Yes

Reviewer #2: Yes

Reviewer #3: Yes

5. Review Comments to the Author

Reviewer #1: The manuscript discusses important issues in nutrition which are currently needed to address many nutrition problems in SSA and provide a mechanism for better approaches. The article registrar (PROSPERO) and use of others have been clearly mapped. Generally, paragraphing of the discussion section(from line 173) is poor, needs some review. Presently, paragraphs are too long and makes the reading difficult.

Reviewer #2: The article meets the requirements proposed by journal and knowledge to the scientific community. The authors fulfilled the necessary methodological requirements. Therefore, I recommend the article for publication.

Reviewer #3: Dear authors,

Thanks for sharing your manuscript for review!

I was delighted to read it and can see it was a lot of work.

I have few comments:

In your background section, please introduce and briefly describe the health psychology perspective that you intend to use for your analysis.

Line 111/2 + 140/141: “Further studies were sought by hand-searching bibliographies of included 112 studies, tracking citations on Google Scholar, and asking experts on the topic”. This is not reflected on the PRISMA chart. Also, chart is figure three in text and figure one in appendix. Please revise for consistency.

Line 132: Data analysis section need further details. Please provide information about how the analysis was done and data synthesised. At present, not clear if this was narrative synthesis. It reads more as qualitative description!

Line 153/4: can you please provide further explanation as to how the finding could be informative?

Table 3: very informative table! Please proofread as still has typos e.g. row 1 “The intervention significant* increased consumption of ..”

Line 173 Anthropometric measurement, please consider summarising this section (and the subsequent results sections) around the findings most pertinent to your aim. At present, you are presenting too many results.

Line 274 issues with references

Discussion: very good concepts introduced, I think it will benefit from reorganisation to communicate points clearer.

6. PLOS authors have the option to publish the peer review history of their article (what does this mean?). If published, this will include your full peer review and any attached files.

**Do you want your identity to be public for this peer review?** For information about this choice, including consent withdrawal, please see our Privacy Policy.

Reviewer #1: **Yes: **Ajike Saratu Omagbemi

Reviewer #2: No

Reviewer #3: No

---

## [Decision Letter · Decision Letter 1]

21 Oct 2022

PGPH-D-22-00527R1

Behaviour change interventions improve maternal and child nutrition in sub-Saharan Africa: a systematic review

Dear Dr. Watson,

Thank you for submitting your manuscript to PLOS Global Public Health. After careful consideration, we feel that it has merit but does not fully meet PLOS Global Public Health’s publication criteria as it currently stands. Therefore, we invite you to submit a revised version of the manuscript that addresses the points raised during the review process.

We look forward to receiving your revised manuscript.

Kind regards,

Joao Tiago da Silva Botelho

Academic Editor

Journal Requirements:

Additional Editor Comments (if provided):

There are some methodological limitations, that can be resolved, yet demand further work.

1. Update of the search.

2. Explain the rationale for not conducting meta-analysis. Whether the variability of results exist or not, SWiM guideline should be careful considered.

3. From my assessment there can be the possibility to perform meta-analysis, particularly using a common effect size, such as ratio of means or cohen's d.

Reviewers' comments:

Reviewer's Responses to Questions

**Comments to the Author**

1. If the authors have adequately addressed your comments raised in a previous round of review and you feel that this manuscript is now acceptable for publication, you may indicate that here to bypass the “Comments to the Author” section, enter your conflict of interest statement in the “Confidential to Editor” section, and submit your "Accept" recommendation.

Reviewer #1: All comments have been addressed

2. Does this manuscript meet PLOS Global Public Health’s publication criteria? Is the manuscript technically sound, and do the data support the conclusions? The manuscript must describe methodologically and ethically rigorous research with conclusions that are appropriately drawn based on the data presented.

Reviewer #1: Yes

3. Has the statistical analysis been performed appropriately and rigorously?

Reviewer #1: N/A

4. Have the authors made all data underlying the findings in their manuscript fully available (please refer to the Data Availability Statement at the start of the manuscript PDF file)?

Reviewer #1: Yes

5. Is the manuscript presented in an intelligible fashion and written in standard English?

Reviewer #1: Yes

6. Review Comments to the Author

Reviewer #1: Corrections have been reflected and article is better written

7. PLOS authors have the option to publish the peer review history of their article (what does this mean?). If published, this will include your full peer review and any attached files.

**Do you want your identity to be public for this peer review?** For information about this choice, including consent withdrawal, please see our Privacy Policy.

Reviewer #1: **Yes: **Ajike Saratu Omagbemi

---

## [Decision Letter · Decision Letter 2]

28 Feb 2023

Behaviour change interventions improve maternal and child nutrition in sub-Saharan Africa: a systematic review

PGPH-D-22-00527R2

Dear Dr Watson,

We are pleased to inform you that your manuscript 'Behaviour change interventions improve maternal and child nutrition in sub-Saharan Africa: a systematic review' has been provisionally accepted for publication in PLOS Global Public Health.

Best regards,

Hasanain Faisal Ghazi, phd

Academic Editor

Reviewer Comments (if any, and for reference):

Reviewer's Responses to Questions

**Comments to the Author**

1. If the authors have adequately addressed your comments raised in a previous round of review and you feel that this manuscript is now acceptable for publication, you may indicate that here to bypass the “Comments to the Author” section, enter your conflict of interest statement in the “Confidential to Editor” section, and submit your "Accept" recommendation.

Reviewer #1: All comments have been addressed

Reviewer #4: All comments have been addressed

2. Does this manuscript meet PLOS Global Public Health’s publication criteria? Is the manuscript technically sound, and do the data support the conclusions? The manuscript must describe methodologically and ethically rigorous research with conclusions that are appropriately drawn based on the data presented.

Reviewer #1: Yes

Reviewer #4: Yes

3. Has the statistical analysis been performed appropriately and rigorously?

Reviewer #1: Yes

Reviewer #4: Yes

4. Have the authors made all data underlying the findings in their manuscript fully available (please refer to the Data Availability Statement at the start of the manuscript PDF file)?

Reviewer #1: Yes

Reviewer #4: Yes

5. Is the manuscript presented in an intelligible fashion and written in standard English?

Reviewer #1: Yes

Reviewer #4: Yes

6. Review Comments to the Author

Reviewer #1: All neccessary improvements made and clearly stated

Reviewer #4: (No Response)

7. PLOS authors have the option to publish the peer review history of their article (what does this mean?). If published, this will include your full peer review and any attached files.

**Do you want your identity to be public for this peer review?** For information about this choice, including consent withdrawal, please see our Privacy Policy.

Reviewer #1: No

Reviewer #4: No
